# A comprehensive temporal patterning gene network in *Drosophila* medulla neuroblasts revealed by single-cell RNA sequencing

Hailun Zhu[1], Sihai Dave Zhao[2,3], Alokananda Ray[1], Yu Zhang[1] & Xin Li [1✉]

During development, neural progenitors are temporally patterned to sequentially generate a variety of neural types. In *Drosophila* neural progenitors called neuroblasts, temporal patterning is regulated by cascades of Temporal Transcription Factors (TTFs). However, known TTFs were mostly identified through candidate approaches and may not be complete. In addition, many fundamental questions remain concerning the TTF cascade initiation, progression, and termination. In this work, we use single-cell RNA sequencing of *Drosophila* medulla neuroblasts of all ages to identify a list of previously unknown TTFs, and experimentally characterize their roles in temporal patterning and neuronal specification. Our study reveals a comprehensive temporal gene network that patterns medulla neuroblasts from start to end. Furthermore, the speed of the cascade progression is regulated by Lola transcription factors expressed in all medulla neuroblasts. Our comprehensive study of the medulla neuroblast temporal cascade illustrates mechanisms that may be conserved in the temporal patterning of neural progenitors.

[1] Department of Cell and Developmental Biology, University of Illinois at Urbana-Champaign, Urbana, IL 61801, USA. [2] Department of Statistics, University of Illinois at Urbana-Champaign, Urbana, IL 61801, USA. [3] Carl R. Woese Institute for Genomic Biology, University of Illinois at Urbana–Champaign, Urbana, IL 61801, USA. ✉email: lixin@illinois.edu

 

The heterogeneity of neural fates builds the foundation for constructing complex neural circuits. Integration of spatial and temporal patterning of neural progenitors allows neural progeny to adopt a spectrum of identities (reviewed in ref. [1–4]). Spatial patterning specifies distinct lineages, whereas within a specific lineage, temporal patterning further expands the neural diversity, as neural progenitors undergo gradual transitions along with self-renewal and give rise to a successive series of neural fates [reviewed in ref. [5,6]]. The first TTF cascade was identified in the embryonic ventral nerve cord, where Hunchback (Hb), Kruppel (Kr), Nubbin/Pdm2 (Pdm), Castor (Cas), and Grainy head (Grh) are sequentially expressed in NBs as they age, and are required for the sequential specification of different neural fates[7–12]. Postembryonic NBs, including larval ventral nerve cord, central brain, and optic lobe NBs, also undergo temporal patterning dependent on TTF cascades and/or opposing temporal gradients of two RNA-binding proteins[13–20]. In addition, the intermediate neural progenitors (INPs) of type II NBs have another temporal patterning axis utilizing a TTF cascade to further expand the neural diversity[21–23]. In vertebrates, there is also accumulating evidence that neural progenitors may undergo transcription-dependent temporal patterning [reviewed in[24,25]]. For example, a number of transcription factors were shown to function in retinal progenitors or cortical progenitors to regulate the temporal specification of neural fates[26–35]. Recently, single-cell transcriptomics studies of retinal, cortical, and spinal cord progenitors revealed age-dependent dynamic changes in transcriptional profiles that are transmitted to their progeny[28,36–38]. These studies together suggest that TTF-dependent temporal patterning may be a general mechanism.

The model system we use to study temporal patterning is the medulla part of the Drosophila optic lobe. During development, a wave of neurogenesis sweeps from medial to lateral in the outer proliferation center (OPC) and sequentially converts symmetrically dividing neuroepithelial cells (NE) into medulla NBs[39–41]. Medulla NBs divide asymmetrically multiple times to make a series of Ganglion Mother Cells (GMCs), which then divide to produce postmitotic progeny. Owing to the spreading of the neurogenesis wave, NBs of different ages and their progeny are orderly aligned on the medial to the lateral spatial axis in the developing larval brain (Fig. 1a). This feature makes medulla a great system to study temporal patterning. Previous studies showed that medulla NBs sequentially express Homothorax (Hth), Klumpfuss (Klu), Eyeless (Ey), Sloppy paired (Slp), Dichaete (D), and Tailless (Tll) as they age[18,19]. Among them, Hth, Ey, Slp, and D are each required for the generation of neurons expressing Brain-specific homeobox (Bsh), Drifter (Dfr), Twin of Eyeless (Toy)/Sox102F, and Ets at 65A (Ets65a), respectively (Supplementary Fig. 1a)[18,19,42]. Loss of Klu causes a NB proliferation defect, precluding the examination of neural fates that require Klu, although overexpression of Klu does lead to the generation of ectopic Runt neurons[19]. Similar to vertebrate retinal and cortical progenitors, medulla NBs switch to gliogenesis at the end of the lineage and then exit the cell cycle[18,43].

In the medulla TTF cascade, there are still fundamental questions remaining. First, the several TTFs identified through candidate antibody screening may not compose the complete TTF sequence. Second, although Ey, Slp, and D are each required for the next TTF to be activated, no cross-regulation was identified among Hth, Klu, and Ey. Thus, it is not known how the Ey->Slp->D->Tll TTF cascade is initiated. Third, it was not clear how the oldest medulla NBs switch to gliogenesis, end the temporal progression, and exit the cell cycle. Finally, an even broader question concerns the regulation of the temporal cascade. As a previous TTF is only necessary but not sufficient to activate the next TTF in medulla NBs[18], additional regulators and molecular mechanisms may be involved to achieve the proper regulation of the cascade.

In this work, we use single-cell RNA sequencing (scRNA-seq) to discover all unknown TTFs and additional regulators, as well as to get a global view of the dynamic temporal patterning process of medulla NBs. ScRNA-seq is increasingly used as an unbiased approach to characterize heterogeneous tissues, including invertebrate and vertebrate nervous systems (reviewed in ref. [44–47]). The Drosophila medulla represents a great system to study temporal patterning using scRNA-seq, because at a single time point during development, we can obtain a continuous population of NBs of all ages, and furthermore, we can use known TTFs to mark the relative neuroblast age and verify the inferred pseudo-time trajectory. Applying scRNA-seq to Drosophila medulla NBs enables us to capture all temporal stages and to reveal the gradual change of neuroblast transcriptome with single-cell-cycle resolution. We report the identification of a list of previously unknown TTFs including SoxNeuro (SoxN), doublesex-Mab related 99B (Dmrt99B), Odd paired (Opa), Earmuff (Erm), Scarecrow (Scro), BarH1, BarH2, and Glial cells missing (Gcm). There are extensive cross-regulations among these TTFs and known TTFs that generally follow the rule that earlier TTFs are required to activate later TTFs and later TTFs repress earlier TTFs. Our study reveals a comprehensive temporal patterning cascade: Hth+SoxN+dmrt99B->Opa->Ey+Erm->Ey+Opa->Slp+Scro->D->B-H1&2->Tll, Gcm, that controls the sequential generation of different neural types by regulating the expression of specific neuronal transcription factors. Finally, Gcm instead of Tll is required for both the transition from neurogenesis to gliogenesis and the cell-cycle exit. Moreover, in pursuit of the mechanism behind the regulation of the temporal cascade, we find that the timely progression of the TTF temporal cascade requires Lola transcription factors expressed in all NBs.

## Results

**ScRNA-seq of *Drosophila* medulla NBs.** To perform single-cell transcriptional profiling of *Drosophila* medulla NBs, we dissected and dissociated larval brains into single-cell suspension and used fluorescence-activated cell sorting (FACS) to enrich medulla NBs (Fig. 1b). Medulla NBs were sorted by the co-expression of two transgenes, *SoxN*Gal4 driving UASRed that is expressed in all medulla cells, and E(spl)mγ::GFP that is expressed in all NBs[40]. We performed two rounds of scRNA-seq of sorted medulla NBs using the 10x Genomics Chromium platform. We combined data from both sequencing experiments into a single analysis by integrating them using Seurat[48]. After quality control and filtering for NBs, our data contained 3074 cells expressing between 261 and 6409 genes, with a median of 3682 expressed genes per cell.

To characterize the developmental states of sequenced NBs, we used Seurat to partition the cells into 15 clusters, which we visualized on two-dimensional uniform manifold approximation and projection (UMAP)[49] plots (Fig. 1c). We reasoned that cell-cycle dynamics likely have biological significance in cycling neural progenitors, therefore for our initial analysis, we chose not to regress out cell-cycle effects. On the UMAP plot, clusters 13, 0, 3, 7, 1, 4, 9, 5, 6, 10, and 2 form a continuous cell stream representing the main body of medulla NBs, because cells in these clusters express neuroblast markers such as Deadpan (Dpn), Miranda (Mira) and E(spl)mgamma-HLH, with the exception that cluster 13 cells have a lower level of Mira. In contrast,

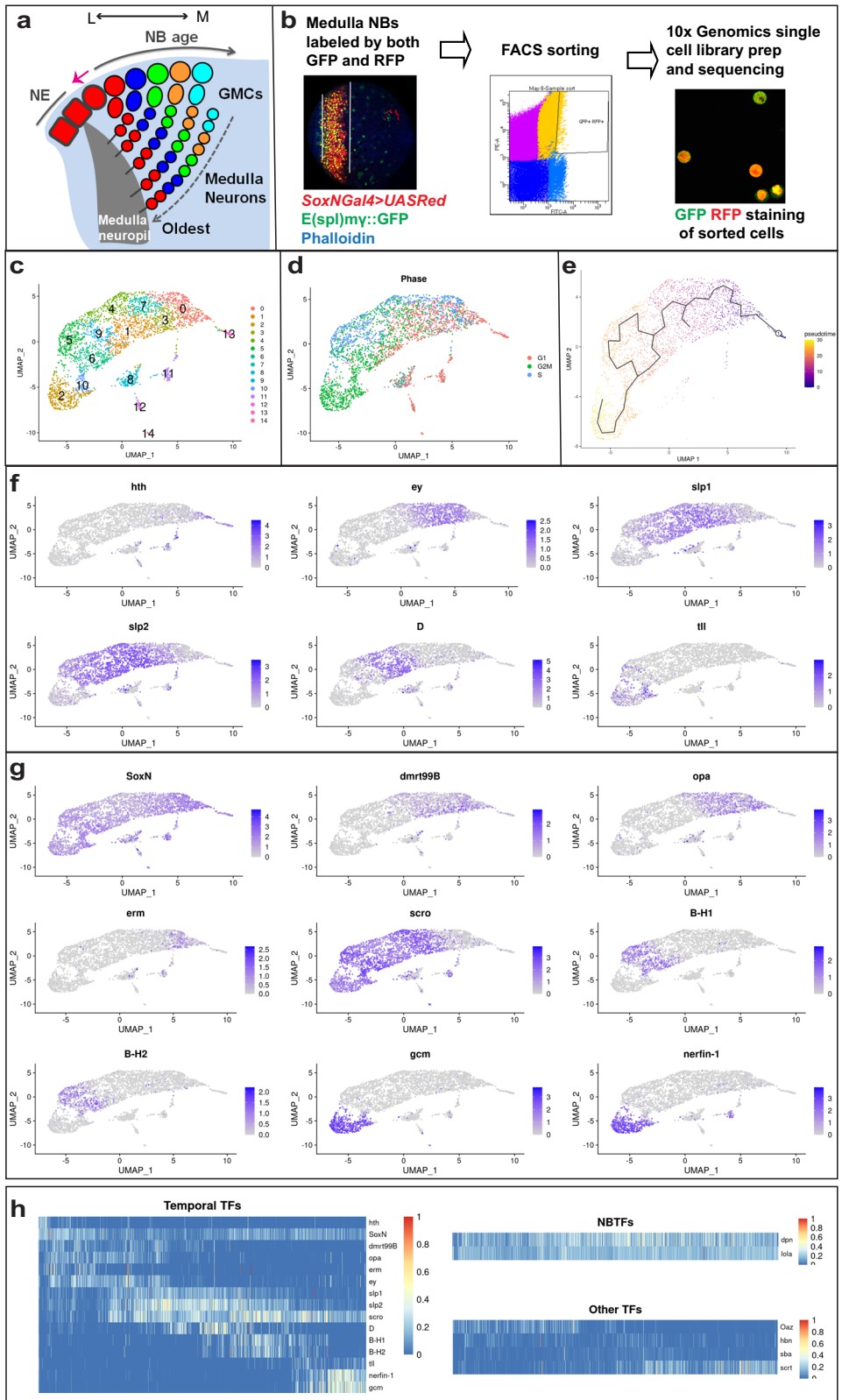

clusters 8, 11, 12, and 14 lie separately from the main body of cells, and most cells in these clusters have an undetectable level of neuroblast markers like Mira (cluster 8, 11, 12, 14) or E(spl) mgamma-HLH (cluster 8, 11, 12) (Supplementary Fig. 1b). Therefore, we regard these clusters as outliers that are not medulla NBs. We also used Seurat to estimate the cell-cycle phase of each cell (Fig. 1d), based on the expression of known

*Drosophila* cell-cycle genes from Tinyatlas at Github[50] (Supplementary Table 1). In the main body of medulla NBs, cluster 13 contains mostly G1 cells, while cluster 2 contains mostly G2/M cells, and in other clusters cells of different phases are mixed. Next, we inferred single-cell pseudotime trajectories using Monocle3[51–53] (Fig. 1e). For this analysis, we removed the outlying clusters 8, 11, 12, and 14. Because Hth is known to be

**Fig. 1 ScRNA-seq of *Drosophila* medulla neuroblasts. a** A schematic drawing of the developing *Drosophila* medulla at the third instar larval stage. A neurogenesis wave (pink arrow) spreads from medial (M) to lateral (L), and sequentially converts NE cells into NBs. Thus, NBs from the youngest to the oldest are aligned on the lateral to medial axis. NBs sequentially express different TTFs, and generate differently fated progeny. The earliest-born neurons of each NB lineage are located closest to the medulla neuropil. The later-born neurons are located at more and more superficial layers. **b** The strategy and workflow of the scRNA-seq of FACS sorted medulla NBs. Medulla NBs are uniquely labeled by the combination of *SoxNGal4>UAS- RedStinger* (red) and *E(spl)mγGFP* (green). This brain is also stained with Phalloidin (blue). Dissected larval brains were dissociated into single-cell suspension and subjected to FACS sorting to enrich medulla NBs. Then 10x V3 Single-Cell libraries were generated using the sorted cells, and sequenced. **c** The sequenced cells were partitioned into 15 clusters using Seurat and visualized on UMAP plots. **d** The estimated cell-cycle phase of each sequenced cell was visualized on UMAP plots. **e** Pseudotime trajectories were generated using Monocle3, with the purple color stands for the earliest pseudotime, and yellow color stands for the latest pseudotime. **f** The expression patterns of known TTFs (Hth, Ey, Slp1, Slp2, D, and Tll) verify the pseudotime trajectories. **g** The expression patterns of newly identified TTFs visualized on UMAP plots. **h** Heatmaps showing the expression levels of three classes of TFs across the pseudotime. Temporal TFs include known and newly identified TTFs; NBTFs include Dpn and Lola that are expressed in NBs of all ages; other TFs include four TFs that show a temporal expression profile, but were not further characterized. The expression levels are visualized as a percentage of the maximum observed expression across all cells.

expressed in the youngest NBs, the root of the trajectory was chosen to be the vertex closest to cells with the highest median hth expression. The inferred developmental pseudotime appears to progress from right to left. Finally, the expression patterns of known TTFs in the medulla NB temporal cascade, which is in the order of Hth, Ey, Slp1/Slp2, D, and Tll, validated the inferred pseudotime trajectory of NBs (Fig. 1f). The Klu mRNA is widely expressed in NBs of all ages, with a relatively lower level in the youngest and oldest NBs, consistent with a recent study showing that Klu expression is activated by Notch signaling in NBs and Klu expression coincides with E(spl)mγ::GFP expression[54]. We noticed that the expression of Klu correlates well with the expression of CycE (Supplementary Fig. 1c), suggesting that it may have a role in neuroblast proliferation consistent with its reported loss of function phenotype[19]. These data confirmed that our scRNA-seq captured a continuous population of NBs of all ages that formed a chronotopic map, with cluster 13 representing the newly transformed NBs at the prolonged G1 phase[55], cluster 0 containing the youngest NBs that have started proliferation, and cluster 2 containing the oldest NBs undergoing the terminal division (mostly at G2/M phases). Other clusters contain cycling NBs in different phases, but there is an overall trend of decreased percentage of cells in G1 phase as NBs age (Supplementary Fig. 1d). To examine whether this is caused by not regressing out cell cycle, we tried regressing out cell-cycle effects for the pseudotime analysis. However, the resulting pseudotime trajectory is not exactly consistent with known TTF expression patterns (Supplementary Fig. 1e–g). Therefore, before visualizing the fraction of cells in different phases of the cell cycle within each cluster, we first manually decided on a cluster ordering that better matched the temporal ordering of known TTFs. The result shows that there does still appear to be some association between pseudotime and the phase distribution of cells, where cells with larger pseudotimes again tend to be in S and G2M phases (Supplementary Fig. 1h). Regressing out cell-cycle effects using an alternative workflow also produced similar results (Supplementary Fig. 1i). Therefore, the observed correlation between the pseudotime and cell-cycle phase distribution is likely to be of biological significance: it is possible that the duration of G1 phase relative to other cell-cycle phases is decreasing as NBs age.

**Two sets of genes show opposite temporal gradients.** Among the genes whose expression changes with pseudotime, there are two sets of genes with high to low or low to high gradients, respectively. Genes encoding ribosomal proteins and metabolism enzymes showed significant high to low gradients (Supplementary Fig. 2a) (Supplementary Table 2), whereas genes involved in gene expression regulation and neural development showed low to high gradients (Supplementary Fig. 2b) (Supplementary

Table 3). These data suggest that as medulla NBs age, protein synthesis and cell growth are gradually downregulated, while differentiation-related genes are upregulated. A similar trend of decreasing transcripts of ribosomal proteins was also observed in vertebrate retinal progenitors[56], suggesting that decreasing protein synthesis is likely a common property of neural progenitors as they age. As there is a change in the distribution of cell-cycle phases as NBs age, we examined whether the gradients still persist if we only examine cells in a certain cell-cycle phase. We repeated our gradient analyses within each cell-cycle phase (Supplementary Fig. 3a–f). In each phase, there was a significant overlap (>50%) between the top 200 genes showing a positive correlation with pseudotime in that phase, and the top 200 genes from our original analysis with all cells (Supplementary Fig. 3g). The same was true for the top 200 genes showing a negative correlation (Supplementary Fig. 3g). These results suggest that the trends we observed correlate with neuroblast age, and are not dependent on the changes in the cell-cycle phase distribution.

**DEG analysis identifies candidate TTFs.** With the neuroblast transcriptome profiles of all ages, we sought to identify more potential TTFs that had temporal expression patterns. Known TTFs were used to mark the relative age of a neuroblast, and to indicate the position where a potential TTF may stand in the temporal cascade. We obtained a pre-compiled list of 755 *Drosophila* TFs[57] identified based on bioinformatic analysis and manual curation. For each non-outlying cell cluster, we performed the DEG (Differentially Expressed Genes) analysis and identified the top 10 TFs in this list that was differentially expressed between the clusters (Supplementary Fig. 2c). Known TTFs are all among these lists. Consistent with the protein expression patterns[18,19], transcripts of "neighboring" TTFs (Ey with Slp, and Slp with D) also have significant overlaps in NBs (Supplementary Fig. 4b–d). However, gaps were observed between Hth and Ey, D and Tll, and after the Tll stage, indicative of missing TTFs (Supplementary Fig. 4a, e). To identify these missing TTFs, we examined the expression of other differentially expressed TFs at the protein level by immunostaining of third instar larval brains using available antibodies or GFP-fusion lines (Supplementary Table 5). To determine whether these TFs play crucial roles in the medulla temporal cascade, we examined cross-regulations between them and the known TTFs. We also assessed whether these TFs are involved in progeny fate specification. After screening through these TFs, we identified SoxN, Dmrt99B, Opa, Erm, Scro, BarH1, BarH2, and Gcm as TTFs in the temporal cascade (Fig. 1g, h). According to the scRNA-seq data (Fig. 1g, h), SoxN transcripts are already present in the newly transformed NBs, remain high in the youngest NBs, and then become reduced from the Ey stage to the D stage and increase again after the D stage. Dmrt99B transcripts are present from the

newly transformed NBs until the Slp stage NBs. The expression of Opa transcripts starts in the youngest NBs before Ey, and continues until the end of the Ey stage, but with a small gap in the middle corresponding to the early Ey stage. Erm transcripts are present at this gap between the two groups of opa-expressing NBs (Supplementary Fig. 5a), i.e., in the early Ey stage NBs. Scro transcripts are present in NBs from about the Slp stage to the final stage. BarH1 and BarH2 transcripts are present in the gap between D and Tll stages (Supplementary Fig. 5b–d), whereas Gcm transcripts are present in the NBs of the final stage, a bit later than the Tll stage (Fig. 1g, h). For a few other candidate TFs with temporal expression patterns, as shown by scRNA-seq, including Oaz, Hbn, Scrt, and Sba (Fig. 1h), we either did not observe any effect on the temporal progression (Oaz, Sba, Scrt) using available RNAi lines (Supplementary Fig. 6a–c'), or we are lacking effective reagents (Hbn). Therefore, we did not include these TFs in further analysis.

**Three NE TTFs specify the first temporal fate.** According to our scRNA-seq data, SoxN transcripts have a rather broad distribution in NBs, with one peak before the Ey stage, and another peak after the D stage (Supplementary Fig. 7a). However, antibody staining showed that SoxN protein is only expressed before the Ey stage corresponding to the first peak (Fig. 2a–a'''), and this discrepancy in mRNA and protein expression patterns suggest that post-transcriptional regulation of SoxN may exist at later stages. Our scRNA-seq analysis did reveal a large number of RNA-binding proteins whose transcript expression patterns correlate with pseudotime (Supplementary Fig. 6d, e). SoxN is a SoxB family HMG-domain transcription factor involved in neuroblast formation and neuron differentiation in the ventral nerve cord[58–61]. Its expression has been noted in the medulla[19], but its function in the medulla has not been studied.

Similar to Hth, SoxN protein expression starts in NE cells before they are transformed into NBs marked by Deadpan (Dpn), and continues in the youngest NBs until the Ey stage (Fig. 2a–a'''). SoxN is also expressed in GMCs and neurons generated before the Ey stage (Supplementary Fig. 7b, b'). To test if SoxN is required for neuron fate specification, we generated SoxN homozygous mutant clones in otherwise heterozygous brains using the MARCM system[62]. First, we observed that SoxN staining was lost in SoxN mutant clones, validating the specificity of the antibody[61] and the mutant (Supplementary Fig. 7c, c'). In the mutant clones, Bsh (brain-specific homeobox) expressing neurons, which are generated in the Hth stage[18,19,63] were lost (Fig. 2b, b'), but Hth expression in NBs was not affected (Supplementary Fig. 7d, d'), suggesting that SoxN cooperates with Hth to specify the Bsh neuron fate. Moreover, with the loss of SoxN, the following temporal cascade still proceeded, but the Slp expression appeared to be slightly accelerated in mutant NBs (Supplementary Fig. 7e–e'''). These results suggest that SoxN is not required for the initiation of the temporal gene cascade in NBs, but it may have a role in the repression of premature Slp expression. We examined whether Hth is required to turn on SoxN. In hth mutant clones, SoxN expression was not affected (Supplementary Fig. 7f, f'). To test if the termination of SoxN is due to the cascade progression, we generated ey-RNAi clones using the ayGal4 (actin>FRT-y+-STOP-FRT-Gal4, in which actin promoter is driving Gal4 expression only after a STOP cassette is excised by the action of hs>FLP) system[64] as well as slp mutant MARCM clones. SoxN expression was expanded into older NBs in ey-RNAi clones but not affected in slp mutant clones, suggesting that Ey inhibits the expression of SoxN (Fig. 2c–c'', Supplementary Fig. 7g, g'). Therefore, SoxN is another TTF that determines the first temporal stage together with Hth. However, neither SoxN nor Hth is required to turn on the expression of later TTFs.

Another TF gene expressed in the youngest NBs is dmrt99B, encoding a DMRT (doublesex- and mab-3-related) transcription factor[65,66]. A Dmrt99B::GFP-fusion protein expressed under genomic BAC endogenous control (from modERN Project) is turned on before the NE to NB transition and remains high in young NBs, and decreases at the beginning of the Slp stage (Fig. 2d–d''). In dmrt99B-RNAi clones generated using the ayGal4 system, the NE to NB transition was not affected, but the neuronal Bsh expression was lost (Fig. 2e–e'', Supplementary Fig. 7i), suggesting that Dmrt99B is also required for the first temporal fate. In contrast to SoxN and Hth, Dmrt99B is required for the later TTF cascade. In dmrt99B-RNAi clones, the expression of the next TTF Opa (see the section below) was lost or dramatically delayed (Fig. 2f, f'), Slp2 was also delayed, and D was lost (Supplementary Fig. 7j–j''). Dmrt99B is not required for the expression of Hth or SoxN in the NBs (Supplementary Fig. 7k–l').

In summary, SoxN, Dmrt99B, and Hth are three TTFs that are first turned on in the NE and all of them are required for the first temporal fate, but only loss of Dmrt99B prevented the progression of the subsequent temporal cascade (Fig. 2g). However, it is possible that partial redundancy may exist within these three TTFs.

**Opa is the link between NE TTFs and the NB TTF cascade.** Opa is a zinc-finger transcription factor homologous to mammalian Zic proteins, and has been shown to function in the embryonic patterning process as a pioneer factor[67,68], as well as in adult head development[69,70]. In the type II neuroblast lineage, Opa is expressed in early-born INPs and required for the temporal progression of the INP temporal cascade[23].

We examined Opa protein expression in the medulla, and observed that it is expressed in two stripes of NBs, consistent with its transcription pattern (Fig. 3a–a'' and Fig. 1g). Opa is not expressed in NE cells, and its expression is activated at the same time as Dpn in the youngest NBs (Fig. 3a–a''). The first Opa stripe is more towards the lateral compared to the Ey stripe (i.e., the first wave of Opa expression in NBs is before Ey), whereas the second stripe is overlapping with the medial half of the Ey stripe (i.e., the second wave of Opa expression is in late Ey stage NBs) (Fig. 3b–b''). The second stripe of Opa expression terminates as Slp expression reaches its peak (Fig. 3c–c''). Inside the medulla, two layers of Opa-expressing neurons, seemingly born from the two Opa windows, were observed. The neurons eventually stop expressing Opa during maturation (Supplementary Fig. 8a–a''').

To test whether Opa is required in the temporal cascade, we used vsxG4, which is initiated in NE cells in the central compartment of the medulla crescent (cOPC)[71] (Supplementary Fig. 8f, h), to drive opa RNAi before the start of its normal expression. In the regions where Opa expression was absent, Ey expression and Slp2 expression were greatly delayed (Fig. 3d, e), which is consistent with what we observed with opa mutant clones (Supplementary Fig. 8b–b''), suggesting that Opa is necessary for the temporal cascade to proceed towards the Ey stage in a timely manner. We also used another regional Gal4, optixG4, which is expressed in the main arms of the medulla crescent (mOPC regions)[71] (Supplementary Fig. 8g, h) to drive opa RNAi, and in the region with opa RNAi, Hth expression was expanded in both NBs and progeny (Fig. 3f, g, g'). Therefore, Opa is required for the repression of the previous TTF Hth, and the timely activation of the next TTF Ey (Fig. 3o).

Next, we examined whether known TTFs regulate opa expression. In ey-RNAi clones, Opa expression persisted without a gap from the youngest NBs to the oldest NBs (Fig. 3h, h'), suggesting that Ey (and/or any later TTF in the temporal cascade) is required to repress Opa in both the gap and the older NBs. In

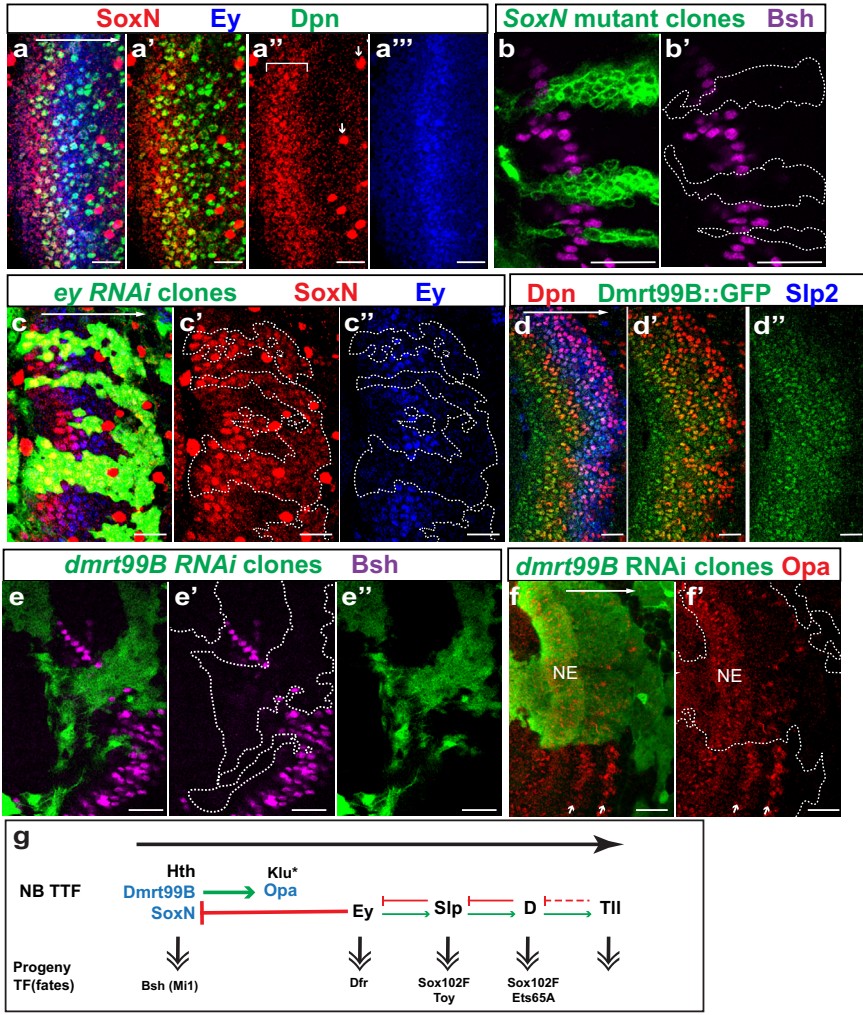

**Fig. 2 SoxN and Dmrt99B are two TTFs that function together with Hth in the first temporal stage.** In all images of this and the following figures, lateral is to the left, and medial is to the right. The large white arrow from left to right indicates the NB age from the youngest to the oldest. White dashed lines indicate clone margin unless otherwise noted. **a–a'''** SoxN protein (red) is expressed before NB formation and in the youngest NBs (marked by Dpn in green) before Ey (blue), and this expression domain is indicated by a white bracket. SoxN is also expressed in glial cells, and examples are indicated by small white arrows. **b, b'** In *SoxN*$^{NC14}$ mutant clones marked by GFP (green), Bsh (magenta) is lost (in 12 out of 12 clones). **c–c''** In *ey-RNAi* clones marked by GFP (green), Ey (blue) is lost, and SoxN (red) expands into older NBs (in 11 out of 11 clones). **d–d''** Dmrt99B::GFP (green) is expressed before NB formation and in the young NBs (NBs marked by Dpn in red) until early Slp (blue) stage. **e–e''** In *Dmrt99B-RNAi* clones marked by GFP (green), Bsh (magenta) is lost (in seven out of seven clones). **f, f'** In *Dmrt99B-RNAi* clones marked by GFP (green), Opa (red) is lost or greatly reduced (16 out of 16 clones). White arrows indicate the two stripes of Opa expression in NBs. **g** A schematic model showing the three NE TTFs and their cross-regulations with NB TTFs. Scale bars: 20 μm.

addition, neurons born at different time points all inherited Opa in the *ey-RNAi* clones. Since the termination of the second stripe of Opa correlates with the upregulation of Slp, and Ey is required to activate Slp, it is possible that the de-repression of Opa in older NBs in *ey-RNAi* clones is owing to loss of Slp. To test whether Slp is required to terminate the second wave of Opa expression, we generated *slp* mutant clones, inside which the expression of the second stripe of Opa was expanded towards the oldest NBs (Fig. 3i, i'). However, the gap between the two stripes was not affected in *slp* mutant clones. Therefore, Ey is required to repress the first stripe of Opa, whereas Slp is required to repress the second stripe of Opa (Fig. 3o).

Finally, we examined whether Opa controls the specification of neural fates. A layer of neurons expressing Runt are born between the Hth stage and the Ey stage[19] (Supplementary Fig. 8c). Runt-expressing neurons were unaffected in *hth* mutant clones, and they were expanded in *ey* mutant clones, suggesting that a TTF repressed by Ey is responsible for the generation of Runt neurons

(Supplementary Fig. 8d, e). Klu was suggested to be the TTF-specifying Runt neurons because overexpression of Klu led to extra Runt neurons[19]. However, Klu and Ey do not regulate the expression of each other, and *klu* mutant caused NB proliferation phenotype, precluding examination of the requirement of Klu for certain neural fates[19]. As the first stripe of Opa is turned on after Hth and before Ey, and we have shown that Ey is required to repress Opa, we examined whether Opa is required to specify Runt neuron fate. In *opa-RNAi* regions, Runt neurons were largely lost (Fig. 3g, g'), suggesting that the first stripe of Opa indeed serves as a TTF between the Hth stage and the Ey stage, and is required for Runt neuron determination (Fig. 3o).

**Cross-regulatory network between Opa, Erm, and Ey.** According to the scRNA-seq data, Erm transcripts are present at the gap between the two groups of Opa-expressing NBs. Erm, an ortholog of mammalian Fezf2, is a zinc-finger transcription factor

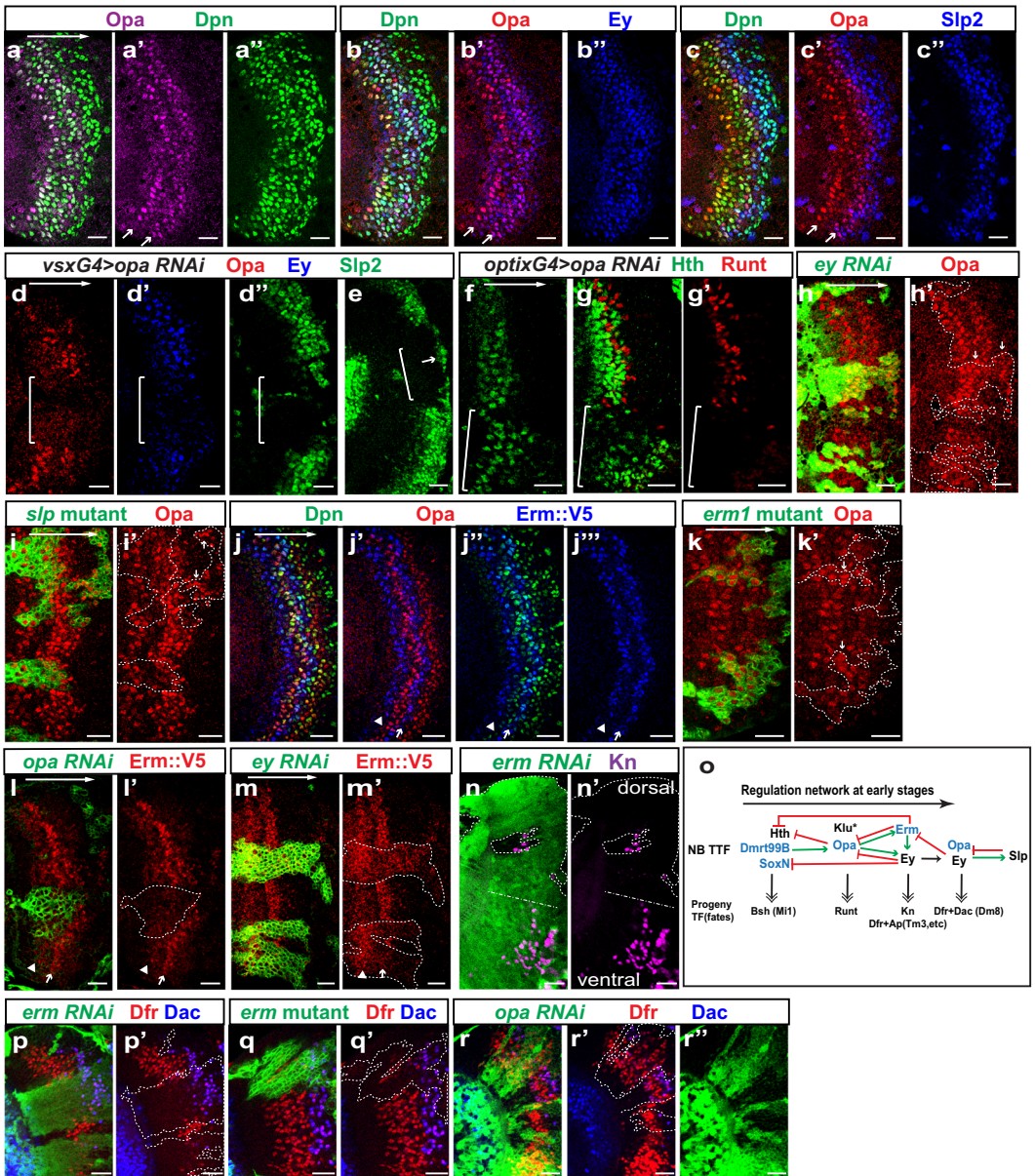

**Fig. 3 Opa and Erm are among the early TTFs. a–a″** Opa protein (magenta) is expressed in two stripes of NBs (marked by Dpn in green). **b–b″** The first stripe of Opa (red) is downregulated as Ey (blue) is upregulated while the second stripe of Opa overlaps with the Ey stripe in NBs (marked by Dpn in green). **c–c″** The second stripe of Opa (red) is downregulated as Slp (blue) is upregulated in NBs (marked by Dpn in green). **d–d″** When *opa-RNAi* is driven by *vsxGal4*, Opa (red) is lost in the center domain (cOPC, white bracket), Ey (blue), and Slp2 (green) are also mostly lost in the same domain (in 11 out of 11 brains). **e** At a slightly deeper focal plane in a *vsxGal4 > opa-RNAi* brain where we can see the oldest NBs, delayed activation of Slp2 (green) can be observed (in 8 out of 11 brains). **f–g′** *opa-RNAi* is driven by *optixGal4* in mOPC indicated by a white bracket (only one side of mOPC is shown in this image). **f** At a surface focal plane, Hth (green) is expanded into older NBs in mOPC (in three out of three brains). **g, g′** At a deeper progeny focal plane, Hth expression (green) is expanded into the later-born progeny, and Runt (red) expressing neurons are lost (in three out of three brains). **h, h′** In *ey-RNAi* clones marked by GFP (green), Opa expression (red) is de-repressed in the gap and also expanded into older NBs (in 15 out of 15 clones). **i, i′** In *slp* mutant clones marked by GFP (green), Opa expression (red) is expanded into older NBs (white arrows) (in eight out of eight clones). **j–j‴** Erm::V5 protein (blue) is expressed in two stripes, one before the NB (marked by Dpn in green) formation (arrowhead), and the other (white arrow) is between the two Opa stripes (red). **k, k′** In *erm1* mutant clones marked by GFP (green), Opa expression (red) is de-repressed in the gap (white arrows) (in 13 out of 13 clones). **l, l′** In *opa-RNAi* clones marked by GFP (green), Erm::V5 expression (red) is lost in NBs (in nine out of nine clones). **m, m′** In *ey-RNAi* clones marked by GFP (green), Erm::V5 expression (red) is expanded into older NBs (in 14 out of 14 clones), but the level is lower than that of the wild-type part of the Erm stripe. **n, n′** In *erm-RNAi* clones marked by GFP (green), Kn (magenta) expressing neurons are lost on the dorsal side (in 15 out of 15 clones), but still present on the ventral side (in 13 out of 14 clones). A white dashed line separates the dorsal vs. ventral side. **o** A schematic showing the regulatory network among early TTFs and the neuron fates generated at each stage. **p, p′** In *erm-RNAi* clones marked by GFP (green), neurons expressing Dfr (red) but not Dac (blue) are lost, and the remaining neurons express both Dfr and Dac thus appear purple (in 13 out of 15 clones). **q, q′** In *erm1* mutant clones marked by GFP (green), neurons expressing Dfr (red) but not Dac (blue) are lost, and the remaining neurons expressing both Dfr and Dac thus appear purple (in 11 out of 13 clones). **r–r″** In *opa-RNAi* clones marked by GFP (green), neurons expressing both Dfr (red) and Dac (blue) are lost, but neurons expressing only Dfr are expanded (in eight out of eight clones). Scale bars: 20 μm.

shown to be expressed in INPs of type II neuroblast lineages where it maintains the INPs' restricted developmental potential[72–74]. The role of Erm in temporal patterning has not been studied before. We used an Erm::V5 line that recapitulates the true expression of Erm[75] to examine its protein expression pattern in the medulla. The staining for V5 marker showed that Erm is indeed expressed at the gap between the two Opa-expressing stripes in NBs, consistent with the transcript pattern as shown by scRNA-seq (arrow in Fig. 3j–j''', Fig. 1g, Supplementary Fig. 5a). In addition, Erm is also inherited in the progeny located between the two layers of Opa+ progeny (Supplementary Fig. 8i). Interestingly, there is another stripe of Erm expressed in NE cells adjacent to the youngest NBs (arrowhead in Fig. 3j–j'''). The expression pattern of Erm in NBs suggests that it may repress Opa expression and generate the gap between the two stripes of Opa. To address this hypothesis, we generated *erm1* mutant clones, and showed that with loss of Erm, the gap in Opa expression in NBs and progeny was no longer present (Fig. 3k, k', Supplementary Fig. 8j, j'), suggesting that Erm does function to repress Opa at the gap, possibly through cooperation with Ey. In *erm* mutant clones, Ey expression became weaker but was still present, and Hth expression was expanded (Supplementary Fig. 8k–l'). In summary, we showed that Erm is activated in NBs at a similar time as Ey, and is required to repress Opa to generate the gap in Opa expression (Fig. 3o).

Next, we examined whether Erm expression is regulated by Opa or Ey using the Erm::V5 line. With the loss of Opa, Erm expression indicated by the V5 marker was greatly lost (Fig. 3l, l'), suggesting that Opa is required for Erm activation. With the loss of Ey, Erm::V5 expression was extended towards older NBs, but interestingly, the expression level seemed to become lower compared to that of its wild-type stripe (Fig. 3m, m'). Therefore, it is possible that a lower level of Ey at the gap between the two stripes of Opa enhances the activation of Erm, but a higher level of Ey and/or a factor-induced by a higher level of Ey at the second Opa window represses Erm (Fig. 3o).

The expression patterns and cross-regulations between Erm, Opa, and Ey suggest that the Ey stage can be subdivided into two sub-temporal stages, with early Ey stage NBs co-expressing Ey and Erm, and late Ey stage NBs co-expressing Ey and the second stripe of Opa (Fig. 1f, g, h, Fig. 3b–b'', j–j'). It is possible that Ey stage NBs generate different neural types in these two sub-temporal stages. To test this hypothesis, we set out to examine whether loss of *erm* or *opa* affects the Ey progeny fates. As previously reported[18], each medulla GMC divides to generate a Notch-on neuron which expresses Apterous (Ap), and a Notch-off neuron which does not express Ap. The Notch-off neurons generated in the Ey stage inherit Ey expression, and they also express a bHLH transcription factor Knot (Kn) (Supplementary Fig. 8m–m"). Ey is required for Kn expression because Kn is lost in *ey-RNAi* clones (Supplementary Fig. 8n, n'). Drifter (Dfr, also known as Vvl) is another transcription factor expressed in the Ey stage progeny, and is lost in *ey* mutant or RNAi clones[18,19]. Dfr expressing neurons can be divided into two large populations. The first population is a layer of Notch-on neurons generated in the early Ey stage that express both Dfr and Ap (Supplementary Fig. 8o–p', Dfr+ cells between the two white dashed lines), and among this population, some neurons also express a weak level of Dac, thus expressing all three TFs (white arrows in Supplementary Fig. 8o–p'). The second population includes clusters of later-born Notch-off neurons that express both Dfr and Dac, but not Ap[18,19,63] (Supplementary Fig. 8o', p', purple cells enclosed by green dashed lines). The Dfr+ Notch-on neurons are specified into several neural types including Tm3, Tm9, Mi10 etc[63]. The Dfr+ Dac+ Notch-off neurons are only generated in certain spatial domains, and they are specified as Dm8 multi-columnar

neurons[63,76]. These Dfr+ Dac+ N-off neurons were still produced in *slp* mutant clones, suggesting that they are born before the Slp stage (Supplementary Fig. 8q, q').

Using these markers, we examined whether Erm is required for the specification of neural fates. In *erm-RNAi* clones, Kn-expressing neurons were always lost on the dorsal side of the medulla, but still present on the ventral side (Fig. 3n, n'); the first population of Dfr expressing neurons were largely lost, but the second population neurons expressing both Dfr and Dac generated at a later stage were still present (Fig. 3p, p'). In *erm* mutant clones, we observed the same phenotype (Fig. 3q, q'). These data suggest that Erm is required for the generation of the first population of Dfr+ neurons in the early Ey stage but not for the second population (Fig. 3o). Erm is also required for the production of Kn-expressing neurons in the dorsal medulla, but Kn-expressing neurons were still present in the ventral medulla with loss of Erm, and a possible reason is that another population of Kn+ neurons not dependent on Erm is generated in the ventral medulla only, possibly in the late Ey stage.

Next, we examined the expression of Kn, Dfr, and Dac in *opa-RNAi* clones. Since Opa is required for Ey expression, and Ey is required for Kn and Dfr expression, we would expect that Kn and Dfr are lost in *opa-RNAi* clones. However, only Kn was lost in *opa-RNAi* clones (Supplementary Fig. 8r, r'), whereas Dfr expression was not lost but even expanded, suggesting that Dfr expression does not require Ey if Opa is not present. However, neurons expressing both Dfr and Dac were lost in *opa-RNAi* clones (Fig. 3r–r"). These together suggest that Opa normally represses the generation of the first population Dfr+ neurons (Dfr+ Ap+ neurons), but is required for the generation of Dfr+ Dac+ neurons (second population). Erm and Ey together are required to turn off Opa at the early Ey stage to allow for the generation of Dfr+ Ap+ neurons. Thus, when Opa is knocked down by RNAi, the repression on the first population of neurons (Dfr+Ap+ neurons) is lifted, and Ey is then no longer required for their generation. After Erm is turned off, the second stripe of Opa together with Ey then promotes the generation of Dfr+ Dac+ Notch-off neurons, possibly acting together with spatial factors. In summary, our data showed that cross-regulatory interactions between Opa, Erm, and Ey subdivide the broad Ey stage into (at least) two sub-temporal stages with Ey/Erm, and Ey/Opa as TTFs, respectively, and these different combinations of TTFs determine different neural fates (Fig. 3o).

**Scro and BarH proteins are TTFs from the middle to late stages.** Based on our scRNA-seq data, scro mRNA is expressed in NBs starting at a similar time as Slp1 and 2. The *scro* gene encodes an NK-2 homeobox transcription factor, the expression of which in the medulla has been indicated by its knock-in mutant alleles in a recent study[77,78].

To study the function of Scro in the temporal cascade, we generated *scro-RNAi* clones, and observed that in such clones, Slp expression was greatly reduced (Fig. 4a, a'), whereas Opa and Ey expression was expanded into older NBs, and D expression was lost (Fig. 4b–d'). This set of data suggest that Scro promotes the transition from the Ey stage to the Slp stage by activating Slp expression. However, overexpression of Scro is not sufficient to increase Slp level or activate Slp at an earlier time point (Supplementary Fig. 9a–a"). Since Slp is required to repress Ey and Opa, as well as to activate D, the expansion of Opa and Ey as well as the absence of D expression in *scro-RNAi* clones are likely owing to the weak Slp expression caused by loss of Scro, but a direct role for Scro also cannot be excluded (Fig. 4j). Next, we tested Scro's effect on the neuron fate generated in the Slp temporal window. Sox102F is a transcription factor expressed in

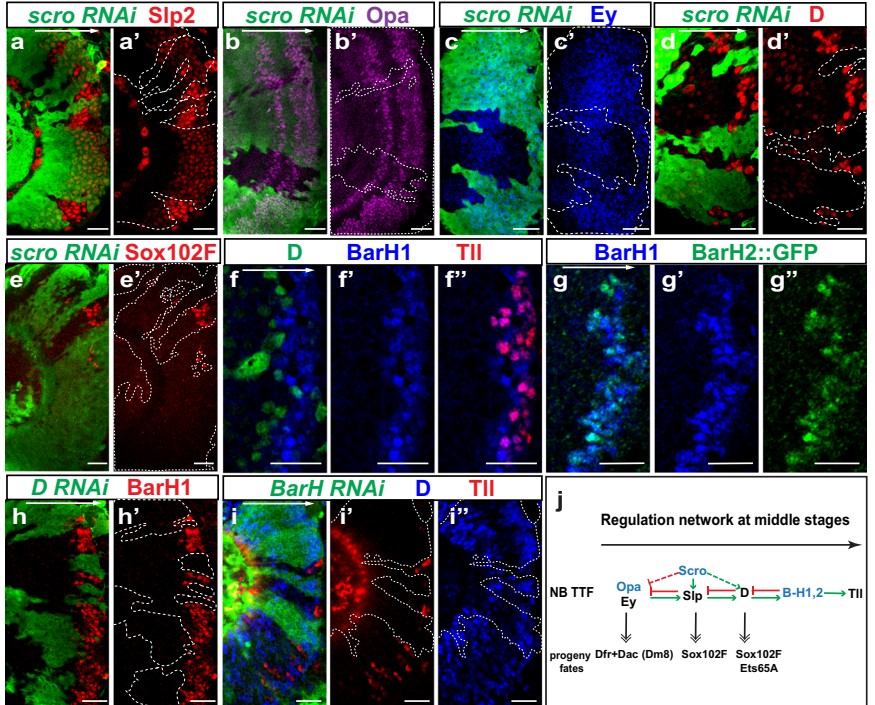

**Fig. 4 Scro and BarH proteins are among late TTFs. a**, **a'** In *scro-RNAi* (BDSC 33890) clones marked by GFP (green), Slp expression (red) is much weaker compared with outside of the clones (nine out of nine clones). **b**, **b'** In *scro-RNAi* (BDSC 33890) clones marked by GFP (green), Opa expression (magenta) is expanded into older NBs (eight out of eight clones). **c**–**d'** In *scro-RNAi* (BDSC 33890) clones marked by GFP (green), Ey expression (blue) is expanded into older NBs, and D expression (red) is lost (in eight out of eight clones). The D antibody we used has unspecific cross-reactivity with other TFs in young neuroblasts (weak staining), but it specifically recognizes D at the D stage, which is demonstrated in Supplementary Fig. 7h, h'. **e**, **e'** In *scro-RNAi* (BDSC 29387) clones marked by GFP (green), Sox102F (red) expressing neurons are not generated (15 out of 17 clones show a complete loss). **f**–**f″** The expression pattern of D (green), BarH1 (blue), and Tll (red) in NBs. **g**–**g″** The expression of BarH1 (blue) and BarH2::GFP (green) in NBs. **h**, **h'** In *D-RNAi* clones marked by GFP in green, BarH1 expression (red) is lost (in nine out of nine clones). **i**–**i″** In *BarH1* and *BarH2* double RNAi clones marked by GFP (green), Tll expression (red) are lost, and D expression (blue) is expanded (in 15 out of 15 clones). **j** A schematic showing the regulatory network among late TTFs: Scro, Slp, D, BarH, Tll, and the neuron fates generated at each stage. Scale bars: 20 μm.

subsets of the neuronal progeny of the Slp stage and D stage NBs (Supplementary Fig. 9b), and it is lost in *slp* mutant clones[42]. In *scro-RNAi* clones, Sox102F[+] neurons were also largely lost, showing that Scro is required for the neuron fate generated in the Slp stage (Fig. 4e, e'). Thus, our data suggest that Scro is required to activate Slp expression to the full level that allows the actual transition to the Slp stage to occur (Fig. 4j). With the loss of Scro, although a weak level of Slp is still expressed in NBs, it is likely not sufficient to specify the correct neural type(s), or promote temporal progression of the cascade. It is also possible that Scro has direct roles in temporal progression in addition to its effect on Slp level.

Another missing TTF at later temporal stages is indicated to be between the D stage and the Tll stage. Consistent with D and Tll antibody staining, our scRNA-seq data also revealed a gap between the D and Tll stages, and showed that *BarH1* and *BarH2* genes are expressed within this gap (Fig. 1g, Supplementary Fig. 5b–d). *BarH1* and *BarH2* are two homologous genes encoding homeobox transcription factors, best known for their function in the development of the eye[79,80]. In the medulla, consistent with our scRNA-seq data, both proteins are expressed in medulla NBs between the D stage and the Tll stage. D, BarH proteins, and Tll are expressed in three consecutive stripes with BarH proteins in the middle (Fig. 4f–f″, Supplementary Fig. 9c–c″). Although D expression and Tll expression do not overlap with each other, the expression of BarH1 and BarH2 overlaps with both D and Tll. BarH1 and BarH2 are initiated at almost the same time, and their stripes in NBs are much overlapped, but the highest level of expression of the two TFs is

achieved at different times in different cells (Fig. 4g–g″). The inequivalence of their expression is amplified in neurons that are born at the BarH stage, as neurons expressing only BarH1 or BarH2 were observed inside the medulla (Supplementary Fig. 9d–d″).

If BarH1 and BarH2 are TTFs between D and Tll, they should be activated by D, and be responsible for terminating D expression and activating Tll expression. We used *ay*G4 to drive D RNAi, and observed loss of BarH1 expression in *D-RNAi* clones (Fig. 4h, h'), suggesting that D is required for BarH1 activation. To test whether BarH1 and BarH2 are required for the transitions from the D stage to the Tll stage, we generated clones in which RNAi of both *BarH* genes was induced. Inside such RNAi clones, Tll was lost while D was expanded to the oldest NBs (Fig. 4i–i″). Next, we tested whether BarH1 and BarH2 are individually required in the transitions. With the loss of BarH1 or BarH2 alone, normal Tll expression was observed in NBs (Supplementary Fig. 9e, g), suggesting that BarH1 and BarH2 act redundantly to activate the next TTF Tll (Fig. 4j).

**Gcm and possibly Nerfin-1 are required at the final stage.** Previously it was thought that Tll is the last TTF expressed in the oldest medulla NBs that produce glia, but whether Tll indeed has a role in gliogenesis has not been examined[18]. Our scRNA-seq data suggest that there is another temporal stage after the Tll stage marked by the expression of Gcm, Nerfin-1, and Dacapo (Dap) (Fig. 1g, Supplementary Fig. 10a). Gcm, a zinc-finger

transcription factor, was shown to be essential for glial fate determination in the embryonic nervous system and larval visual system[81–84]. Then another study showed that Gcm is expressed in a group of precursors located at the border between the optic lobe and the central brain and required to generate medulla neuropil glia (mng)[43]. Our scRNA-seq data suggest that these precursors are the medulla NBs at the final stage, rather than a separate group of dedicated glial precursors.

A Gcm::GFP-BAC line was used to examine the protein expression pattern of Gcm, and it was confirmed that Gcm protein is expressed after Tll in the oldest medulla NBs marked by Dpn and Miranda (Fig. 5a–b"' and Supplementary Fig. 10b–b"'). Some of these Gcm-positive NBs were undergoing mitosis, as shown by strong Anti-phospho-Histone H3 staining (Fig. 5b±b"'). Gcm expression has a significant overlap with Tll, but in a more restricted stripe closer to the central brain. A high level of Gcm expression is often observed in NBs with a reduced level of Tll. Gcm-expressing progeny later activates Repo expression, as suggested by the co-expression of Gcm::GFP and Repo in the migrating mng generated by the oldest medulla NBs (Supplementary Fig. 10c–c"). As mng migrate towards the medulla neuropil, Gcm expression reduces, whereas the expression of Gcm2, a homolog of Gcm, increases (Supplementary Fig. 10d–d"). Gcm2 is not expressed in the NBs[43] (Supplementary Fig. 10d–d"). Although some Gcm-expressing NBs and newly generated mng still transiently retain a low level of Tll, most Tll-expressing progenies do not express Gcm or Repo, and a few of them start to express a low level of Dac (Supplementary Fig. 10c–c" and e–f"), suggesting that Tll is a TTF activated before Gcm, and that Tll stage NBs produce Tll+ neurons but not glia. These neurons appear to only express Tll for a short time. Tll is gradually turned off in mature neurons, whereas in some of them Dac is being turned on.

Thus, Gcm is expressed at the final stage when NBs transit from neurogenesis to gliogenesis and subsequently exit the cell cycle. Dap, a cell-cycle inhibitor orthologous to vertebrate Cdkn1a/P21, is expressed in a similar pattern as Gcm (Fig. 1g, Supplementary Fig. 10a). Tll is expressed in a slightly earlier stage, and loss of Tll does not affect the cell-cycle exit or glia production (Supplementary Fig. 11a–a"). Thus, Gcm may be the critical regulator that functions in promoting gliogenesis and cell-cycle exit.

We examined the function of Gcm by generating gcm mutant clones where both gcm and gcm2 were mutated. In a wild-type brain, cells expressing both Dpn and Tll were not present deep inside the medulla where only neurons and glia are located. However, with loss of gcm/gcm2, ectopic Tll+ Dpn+ cells were observed at deeper focal planes (Fig. 5c–c"). Those cells should be the Tll+ NBs unable to transit into the glia producing and cell-cycle exiting mode, and instead, they remain at the neuron-producing mode and keep producing supernumerary Tll+ Dac+ neurons (Supplementary Fig. 11b–b"). To examine specifically the function of Gcm, we generated gcm-RNAi clones. With loss of Gcm only, we also observed that the oldest Tll+ NBs failed to exit cell cycle and generated excessive Tll+ Dac+ neurons (Fig. 5d–e"' and Supplementary Fig. 11c–c"'). These results suggest that Gcm is required to repress Tll, and to end the temporal progression. The extended NB proliferation could partially be due to the loss of Dap expression in both gcm-RNAi clones and gcm mutant clones (Fig. 5f–f" and Supplementary Fig. 11d–d"). Confirming Gcm's known role in gliogenesis, Repo+ glia around the medulla neuropil were never observed inside gcm-RNAi clones or gcm mutant clones (Supplementary Fig. 11e–e"). To determine if Gcm is sufficient to promote gliogenesis and terminate neuroblast proliferation, we tested the effect of misexpression of Gcm in younger NBs. The clones mis-expressing Gcm were small, and

composed mostly of NBs marked by Dpn and ectopic glia marked by Repo (Supplementary Fig. 11f–f"). Dap was also ectopically activated in Gcm misexpression clones (Fig. 5g–g"), suggesting that Gcm is sufficient to promote gliogenesis and induce Dap expression. In summary, this set of data support that Gcm is the final TTF required for the switch to gliogenesis and for ending the temporal cascade possibly through activating Dap.

Tll is not required for glia production, and this suggests that Tll is not required to activate Gcm. We reasoned that there should be another TTF that is directly upstream of Gcm and required for Gcm expression. Therefore, we tested if BarH genes are the upstream TTFs. Although mng produced by the oldest NBs line up around the medulla neuropil continuously in a wild-type brain (Supplementary Fig. 11g–g"), in BarH1 and BarH2 double RNAi clones, Gcm-GFP expression and mng were lost, suggesting that BarH1 and BarH2 are upstream of Gcm in the temporal cascade (Fig. 5h–h"' and Supplementary Fig. 11h–h"). However, mng production was normal with individual RNAi of BarH1 or BarH2 (Supplementary Fig. 9f, h), suggesting that BarH1 and BarH2 act redundantly to promote the temporal cascade towards the final stage. In summary, BarH1 and BarH2 are required to activate both Tll and Gcm, but Tll is activated slightly before Gcm, and Gcm is then required to repress Tll, and in the meantime promote gliogenesis and cell-cycle exit possibly by activating Dap (Fig. 5q).

Besides Gcm, Nerfin-1 is also expressed in the oldest NBs. Nerfin-1 is a zinc-finger transcription factor expressed in postmitotic neurons and is required for maintaining their differentiated status through inhibition of Notch activity[85–87]. According to our scRNA-seq data, Nerfin-1 transcripts are present in the oldest NBs similar to Gcm transcripts (Fig. 1g). Using a Nerfin-1::GFP-BAC line, we showed that Nerfin-1 protein is expressed mostly in maturing neurons as previously reported (Fig. 5i). However, co-expression of Dpn, Gcm, and Nerfin-1 can be observed in the nuclei of the oldest NBs (Fig. 5i–j', Supplementary Fig. 12c–c"). The oldest medulla NBs express nuclear Prospero (Pros)[18], and we also observed co-localization of nuclear Pros with Nerfin-1::GFP (Supplementary Fig. 12a–b'). Nerfin-1 expression persists in the newborn glia generated by the oldest NBs for a short time and is lost as the glia mature and migrate. Next, we tested whether Nerfin-1 is required for the cell-cycle exit and glia production. We used eyG4 to drive nerfin-1 RNAi, and showed that Nerfin-1 could be another critical regulator for the final-stage NBs. In a wild-type control brain, mng produced from the oldest NBs line up the medulla neuropil continuously (Fig. 5k). In contrast, with loss of Nerfin-1, only a few scattered mng were observed around the medulla neuropil (Fig. 5l) (the reduction is highly significant: $p = 3.49 \times 10^{-9}$ by two-sided $t$ test, $n = 5$ brains each. Source data are provided as a Source Data file). This suggests that gliogenesis is affected by the loss of Nerfin-1, and it is possible that NBs could be stuck at the previous Tll stage. With the loss of Nerfin-1, we also observed numerous ectopic Dpn+ NBs inside the medulla (Fig. 5n), consistent with previous reports showing that loss of Nerfin-1 causes de-differentiation of neurons back into NBs[85–87]. However, if the oldest NBs fail to exit the cell cycle, they can also be among these ectopic NBs. To test this possibility, we examined Tll expression. Tll is transiently expressed in the newly born progeny from Tll+ NBs, and will be lost soon after. In a wild-type brain at a deeper focal plane, NBs are only observed at the surface, and Tll is observed in old NBs and their newly born progeny just below the surface (Fig. 5o arrows). The lineages in the middle of the brain have been completed (NBs have finished generating mng and exited the cell cycle), and thus no NB or Tll+ progeny is observed (Fig. 5m, o). However, with the loss of Nerfin-1, Tll+ progeny continued to be produced throughout the brain (Fig. 5p).

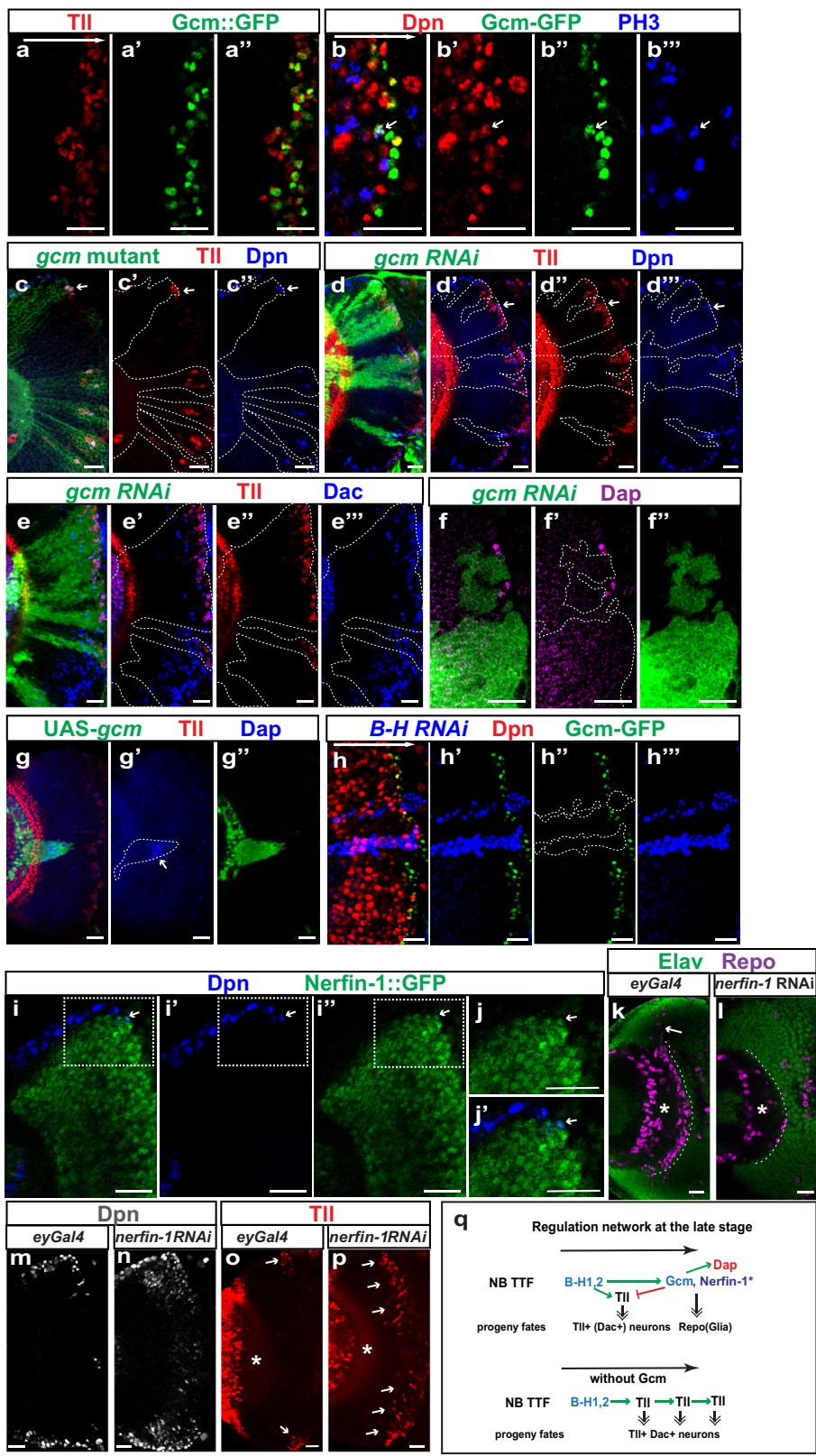

This set of data suggests that Nerfin-1 may also be required in the oldest NBs for the transition from the Tll stage to gliogenesis and for exiting the cell cycle (Fig. 5q). However, another possibility for the loss of glia is that the transient expression of Nerfin-1 in newly born glia is required to prevent them from transforming back to NBs.

Finally, we examined whether Nerfin-1 regulates Gcm expression. With loss of Nefin-1, Gcm-GFP was still expressed in the oldest NBs (Supplementary Fig. 12d–d'''), suggesting that Gcm expression does not depend on Nerfin-1. In summary, at the final stage, Gcm and possibly Nerfin-1 act to promote gliogenesis and cell-cycle exit.

**Fig. 5 The roles of Gcm and Nerfin-1 in termination of the temporal cascade. a–a″** The expression of Tll (red) and Gcm::GFP (green) in NBs. **b–b‴** The staining of Gcm-GFP (green) and PH3 (blue) in NBs marked by Dpn (red). The arrow is pointing at an NB of the final stage going through mitosis, indicated by co-staining for Dpn, Gcm-GFP, and PH3. **c–c″** In *gcm* mutant clones (marked by GFP in green), ectopic NBs marked by Dpn (blue) and Tll (red) are present in a deep progeny focal plane, along with ectopic Tll+ progeny surrounding the ectopic NBs (in 17 out of 17 clones). **d–d‴** In *gcm-RNAi* clones (marked by GFP in green), more NBs marked by Dpn (blue) and Tll (red) are present, along with ectopic Tll+ progeny surrounding the ectopic NBs (in 25 out of 25 clones). **e–e‴** In *gcm-RNAi* clones (marked by GFP in green), the number of Tll (red) and Dac (blue) double-positive cells is increased at a deep progeny focal plane (in 16 out of 16 clones). **f–f″** In *gcm-RNAi* clones (green), Dap (magenta) is lost (in nine out of nine clones). **g–g″** In *gcm* misexpression clones (green), Dap (blue) is ectopically activated (in 10 out of 10 clones). **h–h‴** In *BarH1* and *BarH2* double RNAi clones marked by β-Gal (blue), Gcm-GFP (green) is absent (in five out of five clones). **i–j′** The expression of Dpn (blue) and Nerfin-1::GFP (green) in a cross-sectional view. The rightmost NB is the oldest NB that turns on Nerfin-1 (arrow). **j–j′** A zoomed-in image of the outlined region in **i**. **k, l** In deep progeny focal planes, glia are marked by Repo (magenta), and neurons are marked by Elav (green). The asterisk indicates the center of the medulla neuropil, and the white dashed line indicates the position where mng should be aligned. **k** In *eyGal4* control brains (*n* = 5), mng continuously aligns the medulla neuropil, and the migrating mng stream is indicated by a white arrow. **l** When *Nerfin-1* RNAi is driven by *eyGal4*, only scattered mng is observed (in five out of five brains). **m** At a focal plane slightly deeper than the surface focal plane, most NBs marked by Dpn in an *eyGal4* control brain are located at the surface of the medulla. **n** When *Nerfin-1* RNAi is driven by *eyGal4*, many ectopic NBs marked by Dpn is present inside the medulla (in five out of five brains). **o** At a deep progeny focal plane, Tll+ (red) progeny (white arrows) are only generated around the surface NBs, while no Tll+ cells are observed in the middle of the brain. **p** At a comparably deep progeny focal plane, when *Nerfin-1* RNAi is driven by *eyGal4*, Tll (red) expressing progeny continue to be produced throughout the brain (in six out of six brains). **q** A schematic showing the regulatory network that are crucial for the final-stage NBs. Asterisk for Nerfin-1 indicates that the exact location of Nerfin-1's action is still not certain. Scale bars: 20 µm.

**Lola factors regulate the speed of cascade progression**. Our scRNA-seq analysis enabled us to identify a fairly complete temporal cascade from start to end. How is the speed of the TTF cascade progression regulated and does it involve any other factors that are not TTFs? To address this, we screened through some TFs that are not expressed in a TTF manner but in all NBs, and we found one gene, *longitudinals lacking* (*lola*) that participates in the temporal cascade regulation.

The gene *lola* encodes about 20 isoforms of transcription factors belonging to a Broad-complex, Tramtrack, and Bric-à-brac/poxvirus and zinc-finger (BTB/POZ) family of proteins. The isoforms have a common BTB domain and different Zinc fingers that give each isoform unique DNA binding specificities[88]. The 20 isoforms show diverse expression patterns in the medulla. For example, Lola-F (isoform nomenclature as in ref. [88]) is expressed in all NBs, all GMCs and newly born neurons, but is downregulated quickly to absence as neurons mature (Fig. 6a±a″, Supplementary Fig. 13a–a″). Lola-N is expressed mostly in mature neurons where it is required to maintain the differentiated state[89]. A Lola-T::GFP-BAC line showed weak Lola-T::GFP expression in a similar pattern as that of Lola-F, although with a much earlier activation starting from NE (Fig. 6b, b′). A Lola-K::GFP-BAC line showed that Lola-K is expressed at a high level in mature neurons and NBs, but is not detected in GMCs (Fig. 6c, c′). Therefore, diverse combinations of different isoforms of Lola could have various functions in NBs, GMCs, and neurons. For example, only NBs express both Lola-F and Lola-K, which may act together to regulate NB-specific processes. Using *optix*G4 which is expressed in the mOPC[71], or *ayG4* to drive *lola* RNAi that eliminates all isoforms of Lola, we observed an expansion of Hth expression, a slight expansion of the first stripe of Opa, and delays in the expression of Erm::V5, Ey, the second stripe of Opa, and Slp to increasing extents in NBs, while the proliferation of NBs was not affected much (Fig. 6d–i‴, Supplementary Fig. 13b–c‴). It has been shown that in *lola* null mutant clones, neurons de-differentiate and ectopic NBs can be observed at deep progeny layer in the late larval stage (more pronounced at 96 h after clone induction), and NB tumors are present in the adult optic lobe[89]. In our *lola-RNAi* clones examined at 72 h after clone induction, the ectopic NBs in the progeny layer were mostly located at the medial edge of the clones, which are likely the oldest NBs unable to terminate the neuroblast fate due to the extreme delay of the TTF cascade (Supplementary Fig. 13d–d″). These ectopic NBs were not located near the surface layer where

normal NBs locate, so the examination of the delayed TTF cascade at the surface NB layer was not affected. We used Dpn as a marker to number the NBs from the lateral (the youngest) to the medial (the oldest). By examining which NB each TTF is first activated in, we can roughly determine the activation time of each TTF. In a wild-type brain, Erm, Ey, the second stripe of Opa and Slp are activated in the 2nd/3rd, the 2nd/3rd, the 3rd/4th and the 4th/5th NB, respectively. However, with loss of Lola, the activation of these TTFs started in the 3rd/4th, the 4th/5th, the 5th to 7th, the 7th to 9th NB instead, respectively (Fig. 6j). The 9th NB is the oldest NB that we could accurately count, therefore, TTFs downstream of Slp were not examined. In terms of the duration of TTF expression, we examined the effect of *lola* RNAi on Hth, Erm, Opa, and Ey. In wild-type brains, the stripe of Hth, the stripe of Erm, the first and the second stripe of Opa all consist of only one to two NBs in width. The stripe of Ey consists of three to four NBs in width. However, with loss of Lola, the Hth stripe and the Erm stripe each consisted of two to four NBs in width, the first stripe of Opa consisted of two to three NBs in width, while the second stripe of Opa consisted of no less than four NBs in width (Fig. 6j). The Ey expression was expanded towards the 9th NB (Fig. 6j), which is the oldest NB that we could reliably assess. NBs older than the 9th NB were dislocated into deep layers, which makes it hard to determine their relative age. Therefore, the Slp expressing domain appeared smaller, and it is not clear whether the Slp stage is expanded or not. The mentioned phenotypes suggest that the temporal progression grows slower and slower without Lola, at least in the first four temporal stages. This set of data suggest that Lola inhibits Hth expression and facilitates the normal temporal progression as a speed modulator in NBs (Fig. 6j).

Next, we examined whether neural fates are also affected when TTF cascade is slowed down by loss of Lola. In our *lola-RNAi* clones 72 h after clone induction, we did not observe massive neural de-differentiation into Dpn+ cells (Supplementary Fig. 13d–d″). Therefore, at this time point, we can still examine the initial neural fate specification. With *lola* RNAi driven by *optixG4* or *ayG4*, we observed slightly prolonged production of Bsh neurons, which is consistent with the expansion of Hth in NBs (Supplementary Fig. 13e–e‴), but the level of expansion of Bsh was not as severe as that of Hth. It is possible that the first stripe of Opa or other TFs expressed at young temporal stages still repress the Bsh neuron fate. In addition to the mild change of Bsh, we observed a great reduction of Runt neurons and Kn

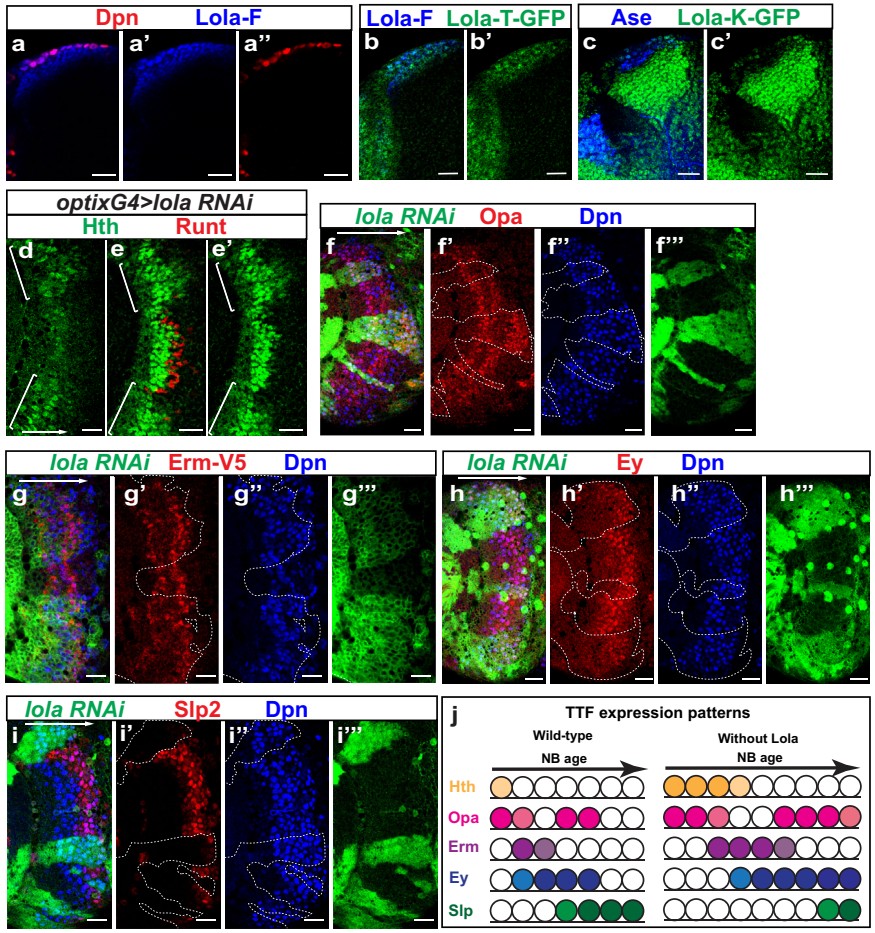

**Fig. 6 Lola regulates the progression of the temporal cascade. a–a″** The expression of Dpn (red) and Lola-F (blue) in a cross-sectional view. **b, b′** The expression of Lola-T::GFP (green) and Lola-F (blue) in a cross-sectional view. **c, c′** The expression of Lola-K::GFP (green) and Ase (blue) marks both NBs and GMCs in a cross-sectional view. **d–e′** *lola* RNAi (BDSC35721) is driven by *optixGal4* in mOPC indicated by white brackets. With *lola* RNAi, Hth (green) is expanded into older NBs (**d**) and later-born progeny (**e, e′**), whereas Runt neurons (red) are mostly lost (**e**) (in five out of five brains). **f–f″′** In *lola-RNAi* clones (VDRC 101925) marked by GFP (green), the second stripe of Opa (red) is activated in the 5th to 7th NB marked by Dpn (blue) (in 19 out of 19 brains), while its activation happens in the 3rd or 4th NB in wild-type regions. **g–g″′** In *lola-RNAi* clones (VDRC 101925), Erm-V5 (red) is activated in the 3rd or 4th NB marked by Dpn(blue) (in 16 out of 16 clones), whereas in wild-type regions, Erm-V5 is activated in the 2nd or 3rd NB. **h–h″′** In *lola-RNAi* clones (VDRC 101925), the activation of Ey (red) is observable in the 4th or 5th NB marked by Dpn (blue) (in 19 out of 19 clones), while wild-type Ey expression starts in the 2nd or 3rd NB. **i–i″′** In *lola-RNAi* clones (VDRC 101925), Slp2 (red) is activated in the 7th to 9th NB marked by Dpn (blue) (in 24 out of 24 clones), whereas in wild-type part of the brains, Slp2 is activated in the 4th or 5th NB. **j** A schematic showing the function of Lola in regulating the temporal cascade. Loss of Lola causes slowing down of the temporal progression. In wild type, only seven NBs were shown because later TTFs were not examined. Scale bars: 20 μm.

neurons, suggesting that Lola proteins are also required for the neuronal fate specification (Fig. 6e, Supplementary Fig. 13e–f″). However, these phenotypes may not be solely the result of losing the isoform combination of Lola in NBs, since in neurons a different combination of Lola isoforms exists and may also be required for neuron fates.

In summary, the requirement of Lola in TTF cascade progression suggests that genes that are expressed in all NBs could also contribute to the regulation of temporal patterning.

## Discussion

Our scRNA-Seq analysis revealed the temporal progression of transcriptional profiles as medulla NBs age at single-cell resolution. We discovered candidates of critical temporal patterning regulators including eight previously unknown TTFs, as well as TFs such as Nerfin-1 and Lola, that are also involved in the temporal patterning process. Further experimental validation of previously unknown TTFs and other crucial regulators confirmed

the accuracy of our high-resolution data, supporting that scRNA-seq is a powerful tool to study the highly dynamic temporal patterning process. Our analysis and further experimental investigation revealed a comprehensive temporal cascade in *Drosophila* medulla NBs: Hth+SoxN+dmrt99B->Opa->Ey +Erm->Ey+Opa->Slp+Scro->D->BarH1&2->Tll, Gcm (Fig. 7b), and also illustrated several principles that are likely conserved during the temporal patterning of neural progenitors.

First, our study identified early temporal factors that initiate the medulla neuroblast TTF cascade. Before this study, Hth was proposed to be the only TTF at play during the earliest temporal stage. Hth is expressed in the neuroepithelium and the youngest NBs. It is necessary for the generation of Bsh neurons, but is required neither for the NE to NB transition nor for the further temporal cascade progression. Loss of Ey also does not affect the termination of Hth[18,19]. These data suggested missing links between Hth and the later TTF cascade. Here, we identified several previously unknown TTFs that linked the whole cascade together. Two of those TTFs that start their expression in the NE,

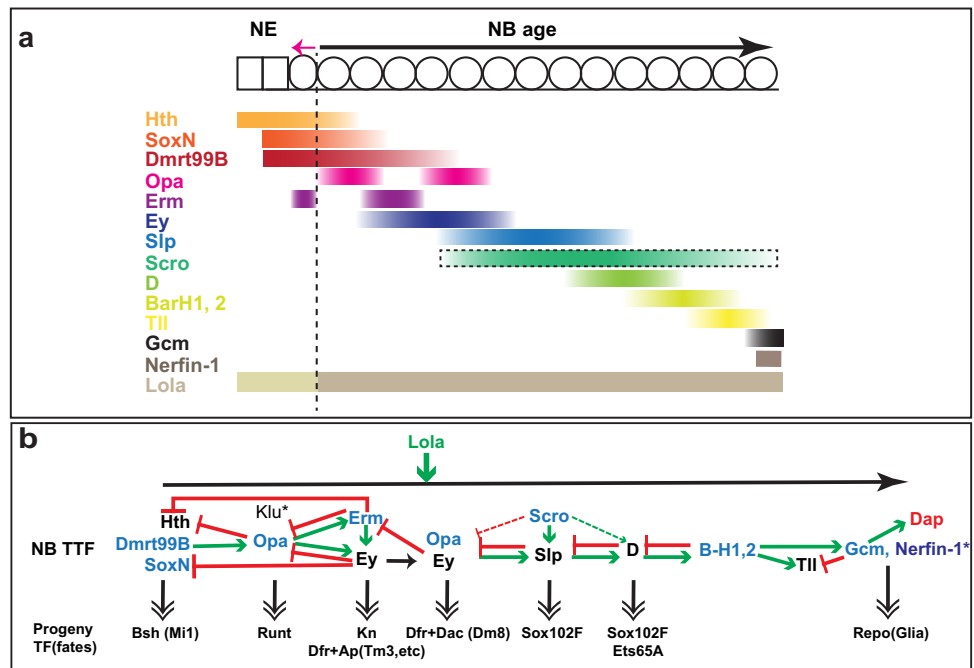

**Fig. 7 A schematic model summarizing the medulla TTF cascade and its regulation. a** A schematic drawing showing the relative expression patterns of the medulla TTFs, Lola, and Nerfin-1. The expression of Scro is indicated only by its transcriptional pattern. The expression of Hth, SoxN, and Dmrt99B all start in NE, Em has a stripe in the transition zone from NE to NB, whereas other TTFs are initiated in NBs. Different isoform compositions of Lola are indicated by different colors. The number of NE and NB cells does not indicate the actual number of cell cycles they go through. **b** A schematic model summarizing the regulation networks of the medulla TTF cascade. Known TTFs are in black, and TTFs identified in this study are in blue. Extensive cross-regulations were identified between these TTFs, which generally follow the rule that a TTF is required to activate the next TTF (green arrows) and repress the previous TTF (red flat-headed arrows), but with a few important exceptions. This TTF cascade controls the sequential generation of different neural types by regulating the expression of neuronal transcription factors, and examples of neural types were also indicated. Note: not all neural fates generated in a certain stage are shown. Lola proteins modulate the speed of temporal progression of the NB TTF cascade. At the final stage, Gcm and possibly Nerfin-1 promote gliogenesis and the cell-cycle exit to end the temporal progression. Cross-regulations are based on mutant phenotypes, and are not necessarily direct regulations.

SoxN, and Dmrt99B, are also required for the first temporal fate (Bsh neurons), and Dmrt99B is required for the timely activation of Opa in the youngest NBs. Opa is then required to activate Ey and repress Hth (Fig. 7a, b). Interestingly, the three TTFs inherited from NE maintain their expression for different durations in NBs, as Hth is repressed by Opa and Erm, SoxN is repressed by Ey, whereas Dmrt99B expression extends until the Slp stage (Fig. 7a, b). Whether this differential downregulation is significant for temporal patterning is currently unknown. However, it is worth noting that the expression of mammalian orthologs of Dmrt99B, Dmrt3, and Dmrta1, also starts in symmetrically dividing early cortical progenitors (NE), and decreases gradually in asymmetrical dividing cortical progenitors due to the direct suppression by FoxG1, the mammalian ortholog of Slp1/2[90,91]. Given the essential role of Dmrt99B in initiating temporal patterning in medulla neuroblast, it will be interesting to investigate whether its mammalian orthologs play conserved roles in the temporal patterning of cortical progenitors.

Second, we showed that a broad temporal stage can be divided into sub-temporal stages by combinations of TTFs, which determine the progeny fates. This is well-illustrated in the Ey stage. The first stripe of Opa is necessary to initiate the expression of Erm and Ey, which are then required to repress Opa in a negative feedback loop, generating a gap in Opa expression. Furthermore, our data suggest that Ey may first enhance the activation of Erm at the gap, but then possibly a higher level of Ey is required to repress Erm, either directly or indirectly. After Erm is turned off, Opa is turned back on. At the same time, Slp has been gradually activated by Ey and Scro, and when it reaches a

certain level, it will repress Opa and Ey to end the Ey stage. Thus, cross-regulations among TTFs divide the Ey stage into (at least) two (sub-)temporal stages determined by the co-expression of Ey and Erm, or Ey and Opa. We showed that different neural types are generated in these two sub-temporal stages, and the first set of neurons require both Ey and Erm, whereas the second set of neurons require both Ey and Opa (Fig. 7a, b). Interestingly to note, the mammalian ortholog of Erm, Fezf2, is also expressed in cortical progenitors and plays important roles in cortical neuron specification[34,35,92].

Third, this study demonstrated that a TTF that is required for the switch to gliogenesis at the final stage is also required for the cell-cycle exit and termination of the medulla TTF cascade. Previously it was thought that Tll stage NBs switch to gliogenesis and then exit the cell cycle, but whether Tll indeed plays a role in these processes has not been studied. Here, our scRNA-Seq data suggested another final temporal stage marked by the expression of Gcm and Dap. Further, we showed that BarH1 and BarH2 are required to activate both Tll and Gcm, but Tll is activated first, and when Gcm is activated, Gcm represses Tll. We showed that Gcm but not Tll is required for the NBs to switch to gliogenesis and exit the cell cycle (Fig. 7a, b). Gcm is well-known for its role in gliogenesis, but here we show that it is also required and sufficient to activate Dap expression in NBs, possibly through which to promote cell-cycle exit and end the temporal progression. In vertebrate retina, scRNA-seq analysis of retinal progenitor cells identified NFI factors as required for both late-born cell fates including Müller glia and for exiting the cell cycle[28]. As neural progenitors often switch to produce glia at the end of the

lineage, it is possibly a general mechanism that factors required for the switch to gliogenesis are also required for the mitotic exit to end the temporal progression.

Another factor that is likely involved in the final stage is Nerfin-1. The expression of Nerfin-1 is observable mostly in maturing neurons, and is required to prevent neurons from de-differentiation[85–87]. However, this TF responsible for maintaining the differentiation status of neurons, is turned on in the final-stage NBs, where it may function to promote gliogenesis and help terminate the temporal cascade on time. The fast exit of the cell cycle at the final stage is likely accomplished because self-renewal repressors that usually function in GMCs and neurons, such as Prospero and Nerfin-1, gather and cooperate in the oldest NBs. Whether Nerfin-1 can be characterized as a TTF is a remaining question. Since Nerfin-1 expression in both the oldest NBs and the newly born glia is very transient, and cell cycle exit is coupled with glia generation in the oldest NBs, it is not easy to distinguish when exactly Nerfin-1 functions to contribute to the termination of the final temporal stage. The mechanism behind Nerfin-1's requirement at the final stage may be different from the mechanism used in neurons preventing their de-differentiation. One evidence is that while a previous study showed that double knockdown of Nerfin-1 and Su(H) could reduce most ectopic NBs generated by single knockdown of Nerfin-1, suggesting that Nerfin-1 represses Notch signaling in neurons to prevent their de-differentiation, there are always several ectopic NBs remaining located at the medial edge inside the double knockdown clones[86]. The location of those ectopic NBs indicates that they are likely the oldest NBs unable to exit the cell cycle. Therefore, Nerfin-1 may function through a different mechanism in the final-stage NBs, which is not dependent on the downregulation of Notch signaling. Finally, we showed that Nerfin-1 is not required for Gcm expression, and it remains to be determined whether Gcm regulates Nerfin-1's expression in this process.

Fourth, we observed complex cross-regulations among TTFs that form temporal gene networks. The model for the cross-regulations between medulla TTFs was that each TTF activates the next TTF and inhibits the previous TTF from the Ey stage to the end of the cascade, exhibiting a simple combination of feedforward activation and feedback repression. However, based on the experimental evidence we produced as well as inferred from the scRNA-seq data, the cross-regulations among TTFs are more complex. One TTF is not necessarily repressed by the very next TTF, or activated by the exactly previous TTF. Hth is repressed by Opa and Erm. SoxN is repressed by Ey, while Dmrt99B is likely to be repressed by Slp or later TTFs. Tll is activated just before Gcm, however, Tll is not required for Gcm's activation. The complexity of their cross-regulation is a way to increase the number of combinations of TTFs in aging NBs, thereby increasing the number of possible neuron fates determined along with the temporal progression. However, the overall trend that early TTFs activate late TTFs, and late TTFs repress early TTFs remains valid.

Finally, we demonstrated that the speed of the TTF cascade progression is regulated by Lola factors expressed in all NBs. Lola proteins belong to a BTB/POZ family of proteins which have been shown to be involved in chromatin remodeling and organization[93]. Certain isoforms of Lola are expressed in all NBs, e.g., Lola-F is activated one cell cycle earlier than Opa. We show that Lola proteins function as a speed modulator of the temporal cascade progression. It represses the expression of Hth, facilitates the activation of Opa and the following TTFs to different extents, thereby guaranteeing a quick transition from the NE TTF network to the NB TTF network. Interestingly, the vertebrate ortholog of *lola*, Zbtb20, was also found to modulate the sequential generation of different neural types in cortical

progenitors[94]. Loss of Zbtb20 causes the temporal transitions to be delayed further and further, very similar to the loss of *lola* phenotype in our system. Thus, it is possible that lola/Zbtb20 play conserved roles in the temporal patterning of neural progenitors.

In summary, the entire life of a medulla neuroblast from the beginning to the end was revealed in this study. Our comprehensive study of the medulla neuroblast temporal cascade illustrated mechanisms that may be conserved in the temporal patterning of neural progenitors. The single-cell RNA-sequencing data provide a plethora of information that allows further exploration of the mechanisms of temporal patterning.

## Methods
### Fly lines and crosses
*Construction of fly lines.* To construct the stock for labeling of medulla NBs for FACS sorting and scRNA-seq, SoxNGal4 (GMR41H10Gal4)[95] was recombined with UAS-RedStinger (BDSC 8547) on Chromosome III, and then crossed with E(spl)myGFP on II[96], to generate the E(spl)myGFP; SoxNGal4 UAS-RedStinger/TM6B stock. To generate SoxN mutant clones, SoxN^{NC14} mutation (BDSC 9938) was recombined onto FRT40A chromosome, to generate the FRT40A SoxN^{NC14} stock.

*MARCM mutant clones with FRT40A mutants.* To generate MARCM mutant clones of SoxN mutant, slp mutant, erm mutant, or gcm gcm2 double mutant, virgin females of yw hs-FLP UASCD8GFP; FRT40A tubGal80; tubGal4/TM6B were crossed with males of FRT40A SoxN^{NC14}/CyO, FRT40A slp^{S37A}/SM6-TM6B (Gift from Andrew Tomlinson[97]), FRT40A erm^1/CyO,GFP (gift from Cheng-Yu Lee[72]), or Df(2 L)200 FRT40A/Gla, Bc (which deletes both gcm and gcm2[84]), respectively. The progeny were grown at 25 °C, heat-shocked once at 37 °C for 40 min at 1st instar larval stage, and then grown at 25 °C for 3 days before dissection of the wandering 3rd instar larvae.

*MARCM mutant clones with FRT82B mutants.* To generate MARCM clones of hth mutant or opa mutant, virgin females of ywhsFLP UASCD8GFP;; tubGal4, FRT82B tubGal80/TM6B were crossed with FRT82B hth^{P2}/TM6B flies (gifts from Richard Mann), FRT82B opa^7 (gift from Deborah Hursh[69]), respectively. The progeny were grown at 25 °C, heat-shocked once at 37 °C for 1 h at 1st instar larval stage, and then grown at 25 °C for 3 days before dissection of the wandering 3rd instar larvae.

*Negatively marked ey mutant clones.* Females of yw, hs-Flp^{1.22};; FRT80B, eyBAC, Ubi-GFP/TM6B,Tb; ey^{J5.71} were crossed to males with genotype hs-Flp^{1.22};; FRT80B; ey^{J5.71}/In(4)ci^D (ref. [18]). The progeny were grown at 25 °C, heat-shocked once at 37 °C for 1 h at first instar larval stage, and then grown at 25 °C for 3 days before dissection of the wandering 3rd instar larvae. Clones in larvae that lacked both GFP fluorescence and staining with an anti-Ey antibody were further analyzed.

*RNAi clones.* RNAi lines used in this study include: UAS-ey-RNAi (BDSC 32486), UAS-dmrt99B-RNAi (BDSC 31982), UAS-opaRNAi (VDRC 101531), UAS-erm^{RNAi} (BDSC 50661), UAS-scro^{RNAi} lines (BDSC 29387, and BDSC 33890 showed the same phenotypes), UAS-D^{RNAi} (VDRC 107194), UAS-BarH1^{RNAi} (VDRC 104681), UAS-BarH2^{RNAi} (VDRC 11570), UAS-tll-miRNA (gift from Tzumin Lee[98]), UAS-gcm^{RNAi} (VDRC 110539, VDRC 2961 showed the same phenotypes), UAS-lola^{RNAi} lines (BDSC 35721, BDSC 26714, VDRC 101925, all showed similar phenotypes), UAS-nerfin-1^{RNAi} (VDRC 101631), UAS-oaz^{RNAi} (VDRC 39214, VDRC 107061), UAS-hbn^{RNAi} (VDRC 103979), UAS-sba^{RNAi} (vdrc 101314), and UAS-scrt^{RNAi} (VDRC 105201). To generate RNAi clones, virgin females of yw hs-FLP; act>y +>Gal4 UAS GFP/CyO; UAS-DCR2/TM6B were crossed with males of each of the RNAi lines. The progeny were grown at 25 °C, heat-shocked once at 37 °C for 8 min at 1st instar larval stage, and transferred to 29 °C for 3 days before dissection of the wandering 3rd instar larvae.

*Region-specific RNAi.* Alternatively, region-specific Gal4s combined with UAS-DCR2 were used to drive RNAi. Virgin females of UAS-Dcr2;Dpn-LacZ/CyO; VsxGal4/TM6B[71] were crossed with males of UAS-opaRNAi (VDRC 101531). Virgin females of UAS-DCR2; optixGal4/CyO[71] were crossed with UAS-lola^{RNAi} lines (BDSC 26714, BDSC 35721). The progeny were grown at 25 °C until 1st instar larval stage and transferred to 29 °C for 3 days before dissection of the wandering third instar larvae.

*Overexpression clones.* UAS-lines used include UAS-Scro-3XHA (FlyORF, F000666), and UAS-Gcm (BDSC 5446). Virgin females of yw hs-FLP; act>y+>Gal4 UAS GFP/CyO were crossed with males of each of the UAS lines. The progeny were grown at 25 °C, heat-shocked once at 37 °C for 8 min at 1st instar larval stage, and transferred to 29 °C for 3 days (UAS-Scro) or 2 days (UAS-Gcm) before dissection of the wandering 3rd instar larvae.

*GFP-BAC and other reporter lines*. Additional lines used in this study include *Dmrt99B::GFP* (BDSC 81280), *Erm::V5* (gift from Cheng-Yu Lee[75]), *ap*[rK568]-*lacZ*[99], *B-H2::GFP* (BDSC 67734), *Gcm::GFP* (BDSC 38647), *gcm-LacZ* (P{PZ}gcm[rA87]/ CyO) (BDSC 5445), *Gcm2::GFP* (BDSC 38646), *lola-T::GFP* (flybase name: *lola.GR-GFP*) (BDSC: 38661), *lola-K::GFP* (flybase name: *lola.I-GFP*) (BDSC: 38662), and *Nerfin-1::GFP* (BDSC 67385).

**Dissociation and FACS sorting of medulla NBs**. For each scRNA-seq experiment, 120 third instar larvae of the genotype *E(spl)myGFP; SoxNGal4 UAS-RedStinger/ TM6B* were washed with PBS twice, and with 70% ethanol for 1 min, and with PBS once again. Each of the brains was dissected on ice in complete Schneider's culture medium (Schneider's Insect medium, supplemented with 10% fetal bovine serum, 2% Pen/Strep, and 0.02 mg/mL insulin). The dissected brains were directly transferred into a glass dish on ice containing Dulbecco's phosphate-buffered saline (DPBS). The dissection was completed within 1 h, and then the supernatant (mainly DPBS) was replaced by 1 mL TrypLE with 1 mg/mL collagenase I and 1 mg/mL papain. The brains were then incubated for 10 min at 30 °C, with gentle shaking at 55 rpm. After removal of the dissociation solution, the brains were carefully washed with complete Schneider's culture medium once and with DPBS twice. The brains were disrupted in 1.4 ml of DPBS with 0.04% bovine serum albumin (BSA) by manual pipetting using a P1000, and then 0.4 ml of DPBS with 0.04% BSA was added to make a total volume of 1.8 mL. The cell suspension was filtered through the cell strainer cap into a 5 mL BDFalcon FACS tube. FACS sorting was done immediately after on BD FACS ARIA II with gentle settings (85 μm nozzle and low pressure of 20 psi). DAPI was added before sorting to distinguish live/dead cells. Among the singlet live cells, GFP and RFP double-positive cells were selected and sorted into DPBS with 0.04% BSA.

For immunohistochemistry of unsorted cells or sorted cells after concentration, the cell suspension was placed onto a coated (poly-D-lysine) dish for 30 min. After fixation with 4% formaldehyde for 10 min, the coverslip was washed four times with PBS. The primary antibodies were incubated for 2 h, and were washed three times with PBS with Tween 20 (PBST). The secondary antibodies were incubated for 30 min, and were washed three times with PBST. The cells were mounted in mounting medium and imaged on Zeiss confocal.

**Construction and sequencing of 10x V3 single-cell libraries**. Single-cell 3′-cDNA libraries were prepared and sequenced at the DNA Services laboratory of the Roy J. Carver Biotechnology Center at the University of Illinois at Urbana-Champaign. FACS sorted cells were immediately concentrated by centrifugation at 500 × g for 5 min, then an additional 800 × g for 5 min to a 40 μl volume. This entire volume was used as input for the 10x library. The single-cell suspension was converted into an individually barcoded cDNA library with the Chromium Next GEM Single-Cell 3′ single-index kit version 3 from 10X Genomics (Pleasanton, CA) following the manufacturer's protocols. The target capture was 10k cells.

Following ds-cDNA synthesis, a sequencing library compatible with Illumina chemistry was constructed. The final library was quantitated on Qubit and the average size was determined on the AATI Fragment Analyzer (Advanced Analytics, Ames, IA). The final library was diluted to 5 nM concentration and further quantitated by qPCR on a Bio-Rad CFX Connect Real-Time System (Bio-Rad Laboratories, Inc. CA).

The final library was sequenced on one lane of an Illumina NovaSeq 6000 S1 flowcell (exp1) or a half lane of an Illumina NovaSeq 6000 S4 flowcell, as paired-reads with 28 cycles for read 1, eight cycles for the index read, and 150 cycles for read 2. Basecalling and demultiplexing of raw data were done with the mkfastq command of the software Cell Ranger 3.1.0 (10x Genomics). Libraries were sequenced to a depth of 2,019,439,522 total reads (1st exp.) and 2,760,057,420 total reads (2nd exp.), corresponding to 6548 cells with a 1821 median UMI counts per cell (1st exp), and 5343 cells with 7508 median UMI counts per cell (2nd exp).

**scRNA-seq analysis**. The sequencing reads were aligned to Ensembl's BDGP6.22 using Cell Ranger (version3.0.1 for 1st experiment, and version 3.1.0 for second experiment) from 10x Genomics, and gene expression levels were counted using Cellranger "Count". In both versions of Cell Ranger, "EmptyDrops" method[100] was used to call cells.

All subsequent analyses were performed in R (version 4.0.3)[101] Quality control on the count data was performed using the package Seurat (version 3.2.3)[48]. To limit the analysis to NBs, only cells expressing Dpn were analyzed. About 47% of cells were excluded. These cells that do not express Dpn also do not express mira, another marker for NBs. Although some of them do express SoxN and E(spl) mgamma, these are expressed at only very low levels. For example, 95% of the dpn-cells show fewer than eight reads of SoxN and 4 reads of E(spl)gamma. Therefore, these cells are not medulla NBs and are excluded from the analysis. Cells were also excluded if 10% or more of their reads came from mitochondrial genes. At this step, 3065 cells were excluded because they had mitochondrial read percentages of 10% or higher, and in these cells, the median percentage was 13.78% and the mean was 16.23%. This left 777 cells from the first experiment and 2302 cells from the second experiment. We also tested setting the mitochondrial read threshold to 20%, but this appeared to introduce too much confounding into the analysis: most of the cells with high mitochondrial percentage cluster together by themselves, and

It is possible that this clustering is driven by confounding factors reflected in the high mitochondrial percentages. Therefore, we kept 10% mitochondrial read threshold.

Data from both experiments were combined into a single analysis. A batch correction was performed using the standard integration workflow in Seurat. Specifically, each dataset was first separately normalized and the top 2000 most variable features were identified. These features were then used to find integration anchors, after which the data were combined using the IntegrateData function. Finally, five outlier cells with large numbers of counts were removed. The remaining 3074 cells were scaled and centered for downstream analysis.

UMAP coordinates were calculated with the RunUMAP function using the top 10 principal components of the 2000 most variable features. Clustering was performed using the FindNeighbors and FindClusters functions with a resolution of 0.9. Cell-cycle scoring was performed using the CellCycleScoring function; see Supplementary Table 1 for lists of the S genes and G2/M genes used. Gene expression levels were visualized with the FeaturePlot function. Differentially expressed genes were identified using the FindAllMarkers function with an FDR cutoff of 0.05.

Developmental trajectories were inferred using the Monocle3 (version 0.2.3.0)[51–53] excluding the outlying clusters 8, 11, 12, and 14. Raw counts data were imported from the integrated Seurat object into a Monocle3 cell dataset object. Preprocessing was performed with the preprocess_cds function and batch correction was performed using the Batchelor algorithm[102] as implemented in the align_cds function. PCA and UMAP embeddings stored in the Monocle3 object were replaced with the corresponding values calculated by Seurat and stored in the Seurat object. Cells were then clustered in Monocle3 using the cluster_cells function and the principal graph was learned using the learn_graph function with the minimal branch length option set to 5. The root node was set to be the vertex closest to cells with the highest median Hth expression. Finally, pseudotimes were calculated using the order_cells function and the trajectory was visualized using the plot_cells function. Genes with temporally patterned expression gradients were identified as those whose expression levels showed a significant Spearman correlation with the inferred pseudotime at an FDR cutoff of 0.05. The top 200 genes with decreasing or increasing gradients, respectively, were analyzed for enriched Go terms for Biological Processes "GOTERM_BP_DIRECT" using the "Functional Annotation Chart" at the DAVID Bioinformatics Resources 6.8 website[103,104] [https://david.ncifcrf.gov/home.jsp].

**Antibodies and immunostaining**. These antibodies are generous gifts from the fly community: Rabbit anti-SoxN (1:100) from Steven Russell[61]; Rabbit anti-Hth (1:500) from Richard Mann; Guinea-pig anti-Run, Rabbit anti-Bsh, Rabbit anti-Slp1, Guinea-pig anti-Slp2, Rabbit anti-D, Guinea-pig anti-Tll, and Rabbit anti-Sox102F (all used at 1:500) from Claude Desplan; Rabbit anti-Zfh1 from Ruth Lehmann, Rabbit anti-D (1:1000) from John R. Nambu[105]; Rat anti-Dfr (1:200) from Makoto Sato[63], Guinea-pig anti-Kn (1:500) from Adrian Moore[106], Guinea-pig anti-Dpn (1:500) from Chris Doe; Rabbit anti-Opa (1:100) from J. Peter Gergen[67]; Rat anti-BarH1 (1:200) from Tiffany Cook[107].

Commercially available antibodies include: sheep anti-GFP (1:500, AbD Serotec, 4745-1051), Chick anti-beta Galactosidase (1:1000, Abcam, ab9361), rabbit anti-RFP (1:1000, Abcam, ab62341), Mouse anti V5-Tag:DyLight®550 (1:200, Bio-Rad, MCA1360D550GA), Rat anti-Deadpan [11D1BC7] (1:200, Abcam, ab195173), Rat anti-Histone H3 (phospho S28) antibody (1:500, Abcam, ab10543). These antibodies are provided by the Developmental Studies Hybridoma Bank (DSHB): mouse anti-eyeless (1:10), mouse anti-Pros (MR1A 1:10), mouse anti-Repo (8D12 anti-Repo 1:50), and mouse anti-Dac (mAbdac2-3, 1:20), mouse anti-Dap (NP1, 1:5), mouse anti-Lola -F (7F1-1D5, 1:20), mouse anti Cut (1:10), mouse anti Broad-core (1:20), and mouse anti Aop (1:10).

Secondary antibodies are from Jackson or Life Technologies (Supplementary Table 4). Immunostaining was done as described[18] with a few modifications: 3rd instar Larval brains were dissected in 1× PBS, and fixed in 4% formaldehyde for 30 min on ice. Brains were washed and then incubated in primary antibody solution overnight at 4 °C, washed three times and incubated in secondary antibody solution overnight at 4 °C, washed three times, and mounted in Slowfade. Images are acquired using a Zeiss Confocal Microscope. Figures are assembled using Photoshop and Illustrator.

**Statistics and reproducibility**. For expression patterns shown in Fig. 2a–a‴, d–d‴; 3a–c″, j–j‴; 4f–g″; 5a–b″, i–j′; 6a–c′ and supplementary Figs. 7b–b′; 8a–a‴, c, f, g, i, m–m‴, o–p′; 9b, c–d″; 10b–e; 11g–g″; 12a–a‴, c–c‴; 13a–a″, at least three brains were imaged for each experiment, and all show the same expression patterns. Three brains are sufficient because wild-type expression patterns are always consistent between different brains.

For loss of function or gain of function experiments, the numbers of animals or clones analyzed are included in each figure's legends. No statistical approach was used to predetermine sample size. Samples sizes were determined following standards in the field and our previous experience. Clonal experiments have internal controls: we compared gene expression in and outside of the clones in the same sample, and only draw conclusions when consistent results are obtained. For quantification of mng number in *eyGal4* and *eyGal4>Nerfin-1 RNAi* brains, mng

marked by Repo were counted on representative focal planes in five brains per genotype, and the *p* value is calculated using a two-sided student's *t* test.

**Reporting summary**. Further information on research design is available in the Nature Research Reporting Summary linked to this article.

## Data availability

The raw and processed scRNA-seq data generated in this study have been deposited in the NCBI's Gene Expression Omnibus database (GEO) under accession code GSE168553. Ensembl's BDGP6.22 is available at [http://sep2019.archive.ensembl.org/Drosophila_melanogaster/Info/Annotation]. Source data are provided with this paper.

## Code availability

Customized codes used for analyzing the scRNA-seq data are available at Zenodo under https://doi.org/10.5281/zenodo.5813627 [https://zenodo.org/record/5813627#.YdUc4WjMJPZ].

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

## Acknowledgements
We thank the Flow Cytometry Facility and the DNA Services Laboratory, and the High-Performance Computing in Biology Group of the Roy J. Carver Biotechnology Center at the University of Illinois at Urbana-Champaign for FACS sorting, and for construction and sequencing of the 10x V3 Single-Cell libraries, and initial analysis of the sequencing data, respectively, and for providing the corresponding method part. We thank the fly community, especially Claude Desplan, Steven Russell, Cheng-Yu Lee, Tzumin Lee, Deborah Hursh, J. Peter Gergen, Tiffany Cook, Louise Cheng, Ruth Lehmann, Adrian Moore, Chris Doe; Richard Mann, Andrew Tomlinson, John R. Nambu, Makoto Sato, and Tetsuya Kojima, for generous gifts of antibodies and fly stocks. We thank the Bloomington *Drosophila* Stock Center, FlyORF, the Vienna *Drosophila* RNAi Center, the Developmental Studies Hybridoma Bank, and TriP at Harvard Medical School (NIH/NIGMS R01-GM084947) for fly stocks and reagents. We would like to thank the NSF-Simons Center for Quantitative Biology at Northwestern University for supporting this project as a Pilot grant, and also the National Eye Institute for grant support (Grant 1 R01 EY026965-01A1 to XL).

## Author contributions
X.L. and H.Z. designed the project and experiments, S.D.Z. analyzed scRNA-seq data, H.Z. and X.L. performed experiments and analyzed data, A.R. and Y.Z. participated in some experiments and data discussion. The manuscript is written by H.Z., X.L., and S.D.Z., with all authors commented.

## Competing interests
The authors declare no competing interests.
