## [Peer Review File · Nature Communications]

A comprehensive temporal patterning gene network in *Drosophila* medulla neuroblasts revealed by single-cell RNA sequencingREVIEWER COMMENTS

Reviewer #1 (Remarks to the Author):

Temporal patterning of the gene regulatory network in developing neural progenitor cells (NPCs) and neurons is an important problem in developmental neurobiology. The OPC of *Drosophila* optic lobe during larval stage is an ideal model system to comprehensively study this issue. Since NEs, NBs, GMCs and neurons sequentially differentiate along the surface of the developing brain, we can examine differential expression patterns of multiple genes in a series of temporal windows during neural development. In this study, the authors combined the scRNAseq technology with molecular genetic analyses in the developing *Drosophila* OPC, and revealed multiple molecular mechanisms that have not been addressed so far. The results of scRNAseq shown in Fig. 1 are striking and will be very useful for those who are studying the same topics. The other chapters describing the results of molecular genetic analyses after L184 are too diverse, although this part is important to validate the results of scRNAseq. The authors should make efforts to clarify the construction of these chapters so that each chapter is dedicated to a clear biological question. Additionally, I would like the authors to address the following points.

Major points:

1. Although Klumpfuss (Klu) was not examined in this paper, its expression has been shown to start after Hth and prior to Ey. So, Klu may cooperate with Opa in NBs. The authors should show the expression profile of Klu in Fig. 1F and discuss its potential function.
2. Line 132: The authors should clarify the reason why the clusters 8, 11, 12 and 14 are regarded as the outliers.
3. L149 'Genes encoding ribosomal proteins andto high gradients': the authors should discuss if this tendency is also applied to the other nervous systems.
4. L169 'According to the scRNAseq data, ...after the D stage': the increase of SoxN expression after the D stage is not found in Figs. 2A and 7A, why?
5. L187 'special post-transcriptional regulation of SoxN exists': the authors should perform in situ hybridization to make this conclusion.
6. L200 'the following temporal cascade still preceded .. (Fig. S2C-D''')': the expression of Slp2 looks slightly accelerated.
7. L230 'in two stripes of NBs': the two stripes of Opa expression are hard to see. The images in Fig. 3A-C should be improved, and quantifications together with Ey and Slp2 are necessary. The two layers of Opa-expressing neurons are unclear in Fig. S3A.
8. L242 'clones are not generated early enough to ...': the clones should be generated in earlier stages to confirm the results of RNAi (Fig. S3B).
9. L285 'consistent with the mRNA pattern (Fig. 3I)': erm mRNA pattern should be shown.
10. L424: the authors should describe if both of gcm1 and gcm2 are mutated.
11. L425-430: cell cycle exit should be directly tested by using general cell cycle markers such as PH3 antibody or BrdU.
12. L436 'but may cooperate with ... (Fig. S5H)': what happens to Dap expression in UAS-gcm clones?
13. L461-485: Southall et al. Dev. Cell 2014 (ref 80) described dedifferentiation of neurons in *lola* mutants. The *lola* RNAi phenotypes in Fig. 6D-G might be caused by neuronal dedifferentiation. The authors should discuss this issue by appropriately citing the reference.

14. L560 'Negative feedback loops with time-delay can generate striped expression patterns': clarify this sentence.

Minor points:

Fig. 1C: the numbers inside the panel are hard to see.

Fig. 5D: Dap staining is hard to see. In general, dark blue is too dark, and should be changed to the other color such as light blue.

Fig. S5E: the overlap between Tll and Dac is unclear.

L73: Klumpfuss

L98 'the gradual change of neuroblast transcriptome'

L106 'but with important exceptions and complexities': this sentence is unclear.

L124 'E(spl)mg::GFP that is expressed in all neuroblasts': a reference is needed.

L141 'Hth, Ey, Slp1/2, D and Tll: according to Introduction, Klu should also be included.

L190: the reference #19 may be wrong.

L250 'Hth expression was expanded in both NBs and progeny (Fig. 3E, F)': is Bsh induced by the expanded Hth expression?

L486: In the Nerfin-1 section, Vissers et al., Cell Rep 25, 1561, 2018 should be appropriately cited.

Reviewer #2 (Remarks to the Author):

Temporal transcription factor-dependent patterning of neural stem cells allows for the generation of a wide variety of differentiated cell types during neurogenesis. This mechanism was first described in *Drosophila* brain and optic lobe neural stem cells. Accumulating evidence suggests that similar mechanisms also occur in vertebrate radial glia. Dr. Li is the first-author on a 2013 Nature paper that describes the temporal cascade of transcription factors that pattern neural stem cells in the *Drosophila* optic lobe. It is clear from Dr. Li's seminal study and the subsequent studies from Dr. Desplan's group that there are yet-to-be-identified genes in this cascade. Dr. Li's group took advantage of the powerful single-cell sequencing technology to define the complete temporal cascade that patterns optic lobe neural stem cells in this study.

In general, I think the scRNA-seq data the authors have generated is very rich and deeply sequenced. The investigators carried out rigorous validation of their findings by examining protein expression and genetic interactions. Aside from adding new genes to the cascade, the overall theme of the temporal cascade remains at the level of genetic interactions, identical to the Li et al., Nature 2013 paper. Perhaps, more in-depth analyses of the scRNA-seq analysis might reveal novel insights into the temporal cascade. For example, Pseudotime analyses shown in Fig. 1C suggest a major branchpoint. Could this branchpoint suggest some sort of heterogeneity among optic lobe neural stem cells and that more than one temporal cascades pattern optic lobe neural stem cells in a region-specific manner?

Major point

1. I think their descriptions of justifications for their in silico filtering methods could be expanded upon to include more descriptions of why cells were excluded (what kinds of cells are they?), and would

Major points.

A. I downloaded the GEO mtx counts table and processed it through MiCV (<https://micv.works>; Michki et al., bioRxiv 2020 (doi: <https://doi.org/10.1101/2020.07.02.184549>)) as follows: Downstream scRNA-seq analysis was performed using MiCV, an interactive web tool that leverages scanpy (Wolf et al., 2018) and other libraries for analysis. In brief, cells were filtered by requiring between 200-15000 unique genes/cell (to exclude debris and some doublets), and genes were filtered by requiring at least 2 cells to express it at greater than 1 UMI/cell. UMI counts were normalized to a total sum of 1000000.0 counts/cell (conversion to counts-per-million/CPM) and subsequently log-transformed by calculating $\ln(1+CPM)$ for each gene for each cell. The top 2000 highly variable genes were identified using the cell_ranger method (Zheng et al., 2017) and these

genes were used to perform a principal component analysis (PCA, n=50pcs). Whenever batch correction was required, data were integrated using the harmony algorithm. A neighborhood embedding was calculated with k=20 neighbors before finally embedding cells in a UMAP projection.

Questions: Why were the other cells (top-right portion of these UMAPs) excluded from the analysis? Are they the high mito-percentage cells? If so, if we consider that 10% may be too restrictive of a filtering step (there is debate around the percentage that indicates poor cell-health, and 20% is not an unheard of cutoff), I wonder a) what the mean/median percentage of mito reads is in this excluded population, and b) if these cells are included regardless of mito-percentage, what (if any) conclusions might change about the proposed TTF progression? Is it possible that this excluded group of cells (or even some groups in the originally included set of cells) represent a separate branch in the TTF progression cascade? Are they actually all NBs? Importantly, do they come from both experimental replicates, or only one of them? From my analysis here, it seems that MB_1 (the first replicate on GEO) is overrepresented in this excluded population, and that MB_2 (the second replicate) makes up the majority fraction of cells in the data included by the authors. Can the authors propose a reason for this distribution? Do they see such a batch-variation in their own analysis, or is this a limitation of the batch-correction method used here? (Harmony, from Korsunsky et. al., 2019, Nature Methods)

2. line 131-133: Cluster 8, 11, 12, and 14 appear to be outliers, but by what criteria other than UMAP position? The rationale used to exclude those cells can be improved (the marker genes expressed by those cells vs. the cells they kept should be briefly described and used for that rationale - UMAP position alone can be an indicator, but in general is not a strong enough criteria).
3. line 137-145: Pseudotime analyses typically require a starting cell to be selected. How was this cell chosen? (typically, by high expression of 1-3 specific genes that are known to be expressed in the starting cell population; I don't believe the authors describe their criteria).
4. line 133-145 and Fig. 1C-D: NBs divide many times throughout their lifetime, going through the entire cell cycle each time. This is not a criticism of their analysis. The authors could attempt to regress out the effect of cell-cycle genes and re-cluster to see if their 'temporal progression' patterns were linked to progression through the cell cycle and not NB age? Alternatively, providing data that shows the fraction of cells from each cluster that are in the G1, G2/M, and S phases (as bar graphs, for instance) may help readers better understand how well-represented each phase is in each NB-type cluster/portion of the pseudotime trajectory. I believe doing a regression/showing this information will actually strengthen the conclusions that authors have made
5. line 184-188 and Fig. 1H: the authors used a SoxN-Gal4 line to drive expression of the reporter they used to physically enrich their cells. Is it surprising that SoxN is broadly expressed in the cells at the RNA-level? I don't think they have compelling evidence that SoxN is 'temporally varying' either, based on their RNA-seq data.
6. The authors propose that three temporal TFs (Dmrt99B, Hth, and SoxN) may function in parallel. Could the author exclude the possibility that Dmrt99B functions upstream of Hth and SoxN? Please examine if Dmrt99B RNAi decreases SoxN and Hth expression.
7. Is Erm expression affected in ey-mutant clones? How about Ey expression in erm-mutant clones?
8. Opa expression patterns in erm clones appear different from ey clones in Fig. 3G & J. There appears to be one wave of Opa expression in erm-mutant clones but two waves of Opa expression in ey-mutant clones. Please clarify. If this is the case, does this result suggest Erm and Ey have different function during this transition?
9. The authors conclude that Scro promotes the transition from the Ey stage to the Slp stage by activating Slp1 expression. Please examine Scro protein expression pattern.

0. Does scroRNAi affect Erm expression?

1. The conclusion for Fig. 5H is overstated because the authors did not include a rescue experiment!. They should perform experiments to demonstrate Tll and Dap are the functional downstream of Gcm in regulation of gliogenesis and cell cycle exit. Otherwise, the authors should tone down their conclusion.

2. Other types of neuroblast showed nuclear Pros in the final division before entering quiescence or terminating the cell cycle. Multiple factors contribute to the cell cycle exit of the old neuroblast. Did you examine if Nerfin-1-positive neuroblasts show nuclear Pros in interphase?

3. Identification of Nerfin-1 in regulating progression and termination of the temporal cascade is interesting. Can Nerfin-1 expressing neuroblasts divide? Could they be GMCs instead? If they divide, is the division symmetric or asymmetric?

4. The authors suggest that Erm represses Opa and that Opa represses Hth. Is there any evidence that this repression cascade is direct?

5. The authors suggest that Scro activates Slp. Does Scro overexpression induce higher Slp expression? Please explain the outcome of this experiment.

16. The authors suggest that BarH1 and BarH2 activate Tll and Gcm, and that Gcm represses Tll. The rationale regarding how BarH1 and BarH2 activate gene expression with different kinetics is very unclear and difficult to imagine. Please clarify.

17. Data in the figure are not trivial for general audience to evaluate because of the amount of background information needed to interpret the staining patterns and the complexity of tissue architecture.

A. Please consider adding a summary illustration describing findings from previous studies and ending each figure by inserting newly obtained information into the illustration.

B. Please consider labeling distinct cell types and distinct waves of gene expression in the figures with arrow, arrowheads, etc, and spatially emphasize where the authors would like the reader to focus on.

C. Please also include the expression pattern of Gal4 drivers used in the experiment in the illustration.

D. It is very difficult to evaluate the boundary of gene expression in multi-color images in many figures. The authors should consider including some single-color images with labeling. In Fig. 2A for example, it is very difficult to see the boundary of Ey-negative stage and -positive stage.

Minor point

1. Please outline clones in 3K', 4B', 4E' 5B' 5C', 5C'', 5D' 5E'', and 5G''.

2. in Fig1B, myGFP?

3. Please clarify the rationale why Dfr expression does not require Ey if Opa is not present.

4. line 23-24, "All ages" could be clearer - this scRNA-seq experiment was only performed at 1 developmental time-point.

5. line 21-22, The authors say it is not known if why 'temporal progression' only happens in NBs and not in progeny. I wonder what 'temporal progression' they would expect in terminally differentiated and non-amplifying progeny, and if they mean just in MNBs or in all NBs. They point out the temporal progression of type-II INPs in [52-53].

6. In Fig1B, what is the blue signal in the image of the larval brain lobe?

7. Is SoxN expression affected in hth-mutant neuroblasts?

8. line 203, what is the ayGal4 system?

Reviewer #3 (Remarks to the Author):

main strength

This study led by Dr. Xin Li is beautifully done and very thorough! It reveals a full set of temporal transcription factors with detailed characterization of TTF cascading, TTFs' roles in neuronal diversification, and regulation of the TTF cascade by additional factors. It provides a complete mechanistic understanding on how a single neural progenitor can consistently yield a series of multiple neuron types. This important and comprehensive work promises to be a classical paper for the field of developmental neurobiology.

weakness of the study

There are no major weaknesses to this study. However, there are a few concepts that could be explained or framed differently or perhaps explored more deeply.

1. Lines 21-22 "It is also not known why temporal progression only happens in neuroblasts but not in their differentiated progeny."

and Lines 84-88: "Furthermore, temporal cascade progression only happens in neuroblasts but not in their differentiated progeny, which makes us to question whether genes functioning in the differentiation axis (NBs -> GMCs -> neurons) might play a role in regulating the temporal progression, which has never been studied before."

This premise doesn't make much sense to me. Temporal progression needs to happen only in the NB. Do you mean to say something along the lines of "It is also not known how progression of the temporal transcription factor cascade is confined only to the neuroblasts. The progeny inherit the TTF of their parent NB and do not express the subsequent TTF in the cascade."

Temporal progression occurring in the post-mitotic progeny wouldn't make sense biologically (though expression of a subsequent neuronal transcription factor could be via a similar mechanism, though it would not be considered temporal progression). Thus, it is not clear what question is trying to be addressed regarding the differentiation axis regulating temporal progression.

2. Lola as speed modulator:

While it is clear that Lola RNAi expands Hth expression domain, the remainder of the TTFs appear to be delayed. However, the data in Figure 6 does not clearly show expansion of the other TTF domains as 6P suggests. Using Dpn as a NB marker and counting the number of NBs expressing each TTF (comparing Lola RNAi and WT) would help make this conclusion more solid.

3. Is Nerfin-1 actually a TTF?

Nerfin-1 in the context that is being examined is not really "functioning along the differentiation axis," rather Nerfin-1 is expressed in the NB. Indeed, Nerfin-1 may be the final TTF that drives the transition from a NB to Glioblast and cell cycle exit and. Figure 6O, suggests that Nerfin is required to turn off Tll. Do Nerfin-1 and Gcm work together (partially redundant) or is there regulation between Gcm and Nerfin-1? It would be interesting to explore the relationship between Gcm and Nerfin-1. Is Nerfin-1 required for Gcm expression or vice versa. Do BarH1/BarH2 turn on Nerfin-1? The Gcm mutant seems to have a milder phenotype than nerfin-1 RNAi with regards to excess Tll positive cells. The # of NBs that did not exit the cell cycle (rather than dedifferentiated neurons) is hard to determine based on Fig6M, perhaps a NB specific driver is needed for Nerfin-RNAi

minor comments:

1. Line 36: When you say "neural stem cells" I assume that you mean vertebrate, mammalian or human neural stem cells.
2. Lines 50-51: "Postembryonic NBs, including larval ventral nerve cord, central brain and optic lobe NBs, also employ temporal cascades that are yet different from the embryonic one." "Temporal cascades" may be too broad...postembryonic CB NBs don't really employ a "cascade" for temporal patterning, as "cascade" implies a series of TTFs driving the subsequent TTF expression.
3. Lines 71-72: "Among them, Hth, Ey, Slp and D are each required for the expression of the corresponding neuronal transcription factors to control neural fates." This is a little bit vague as written—can you clarify what is meant by "corresponding neuronal transcription factors" and "neural fates"?
4. Lines 82-83: "As a previous TTF is only necessary but not sufficient to activate the next TTF in medulla NBs" This information needs a reference
5. Lines 98-99 "We observed growth and metabolism related genes showed high to low gradients". This is grammatically awkward. There are 2 subjects and 2 verbs "we observed" and "genes showed"
6. Figure 1B: FACS sorting: It looks like the PE-A laser settings may have been too high (there seem to be many cells above 105) and many Red/Green double+ neuroblasts may have been missed. Did you also confirm NBs were larger (forward scatter)?
7. Lines 127-128: "After quality control and filtering for neuroblasts, our data contained 3074 cells expressing between 261 and 6409 genes, with a median of 3682 expressed genes per cell." According to the methods (lines 730-731) you sequenced over 11K cells. Were most cells lost during QC or filtering for Dpn? What % of sequenced cells were excluded with QC or the NB filtering step?
8. Cluster 8 seems to be rather large for an outlier. Is there any clue what the "outlier" clusters could be? Could they be NBs from a different brain region? Or a mixture of cell types (rather than single cells).
9. Figure 1F: There is a lot of overlap in the expression of the known TTFs. Does this RNAseq analysis align well with the previously published antibody data? It would be helpful to see a figure demonstrating some of the overlap in expression of TTFs. This would also help show where a TTF may be missing—for instance it appears there are cells in cluster 0 that do not express hth or ey and there seems to be a gap between D and tll expression clusters 5&6.
10. Lines 149-151: references to figure are missing "Genes encoding ribosomal proteins and metabolism enzymes showed significant high to low gradients (Figure S1A), while genes involved in gene expression regulation and neural development showed low to high gradients (Figure S1B)".
11. Do the genes that show opposite expression gradients (lines 147-154 and Fig S1) correlate with cell cycle rather than simply temporal patterning? This is important to consider as the ratio of cells captured in different cell cycle stages correlates with temporal patterning and many of the GO terms are related to cell cycle processes. If controlled for cell cycle stage, would the gradients still persist? For instance, if only cells that were in G1 were assessed over pseudo time, would the expression gradients remain?
12. Lines 163-165 "We next examined the expression of these TFs at the protein level by immunostaining of 3rd instar larval brains using available antibodies or GFP-fusion lines." This data is not shown. It would be helpful to have a table that shows the antibodies/GFP-fusion lines etc. that were tested, and which showed temporal differences.

13. Lines 179-182 "For a few other candidate TFs with temporal expression patterns, including Oaz, Hbn, Scrt and Sba (Figure 1H), we either did not observe an effect on the temporal progression (Oaz, Sba, Scrt) using available RNAi lines, or we are lacking effective reagents (Hbn). Therefore, we did not include these TFs in further analysis."

This data is not shown. Can it be included in the supplement.

14. The fact that SoxN protein has a clear temporal pattern but the transcript less so (temporal post transcriptional regulation) suggests that other potential TTF's transcripts may not be differentially expressed. Are there temporally regulated RNA binding proteins?

15. Since SoxN is post-transcriptionally regulated, the data suggests that Ey (or an Ey downstream gene) is responsible for the post-transcriptional regulation of SoxN.

16. Lines 197-200: The conclusion that SoxN "cooperates with Hth to specify the Bsh neuron fate" should come after Hth expression is described as "unaffected" in SoxN mutants

17. Figure 3 A-C' It is clear that all of these images are from the same sample. The colocalization of red and blue are not as clear as with red/magenta and green. For B and C, it would be helpful to have 3 color panels with Dpn shown to highlight the NBs.

18. The alternating stipes of Erm and Opa proteins are very interesting, suggesting potential cross regulation. Does Erm inhibit Opa expression in the neural epithelial cells? Does Opa inhibit Erm? I later discovered that this was covered in the discussion. Perhaps this sentence (lines 557-558) "Due to the lack of Erm antibody, we haven't examined how Erm expression is regulated" could be moved to the results section.

19. Interestingly, Erm and Opa transcripts do not appear to have such clear alternating expression as the antibody/transgene. Could there be post-transcriptional regulation at play?

20. Figure 6H: Can Dpn, Nerfin and Gcm triple staining be performed? Are there antibodies available for Nerfin-1 or Gcm?

Reviewed by Tzumin & Rosa

Reviewer #1 (Remarks to the Author):

Temporal patterning of the gene regulatory network in developing neural progenitor cells (NPCs) and neurons is an important problem in developmental neurobiology. The OPC of *Drosophila* optic lobe during larval stage is an ideal model system to comprehensively study this issue. Since NEs, NBs, GMCs and neurons sequentially differentiate along the surface of the developing brain, we can examine differential expression patterns of multiple genes in a series of temporal windows during neural development. In this study, the authors combined the scRNAseq technology with molecular genetic analyses in the developing *Drosophila* OPC, and revealed multiple molecular mechanisms that have not been addressed so far. The results of scRNAseq shown in Fig. 1 are striking and will be very useful for those who are studying the same topics. The other chapters describing the results of molecular genetic analyses after L184 are too diverse, although this part is important to validate the results of scRNAseq. The authors should make efforts to clarify the construction of these chapters so that each chapter is dedicated to a clear biological question. Additionally, I would like the authors to address the following points.

Thanks for the reviewer's comments. We have modified the construction of the chapters so that each chapter is dedicated to a clear biological question, which is reflected in the session title.

Major points:

1. Although Klumpfuss (Klu) was not examined in this paper, its expression has been shown to start after Hth and prior to Ey. So, Klu may cooperate with Opa in NBs. The authors should show the expression profile of Klu in Fig. 1F and discuss its potential function.

From our scRNA-seq data, Klu mRNA is widely expressed in neuroblasts of all ages, with relatively lower levels in the youngest and oldest neuroblasts. We have added the expression profile of Klu in Supplementary Figure S1C. The expression of Klu correlates well with the expression of CycE (Supplementary Figure S1C), suggesting that it might be involved in neuroblast proliferation. Consistently, loss of Klu caused neuroblast proliferation defect (Suzuki *et al.*, 2013). However, Klu might still have another role in the specification of neurons, because overexpression of Klu is sufficient to cause expansion of Runt+ neurons (Suzuki *et al.*, 2013). We added these discussions to the text.

2. Line 132: The authors should clarify the reason why the clusters 8, 11, 12 and 14 are regarded as the outliers.

Although clusters 8, 11, 12, and 14 have Dpn expression, most cells in these clusters have undetectable level of other neuroblast markers like Mirander (Mira) (cluster 8, 11, 12,14) or E(spl)mgamma-HLH (cluster 8, 11, 12). Not all Dpn expressing cells are neuroblasts: some lamina precursor cells and some neuroepithelial cells also express Dpn. Therefore, we regard these clusters as outliers that are not medulla neuroblasts. These have been included in Supplementary Figure 1B.

3. L149 ‘Genes encoding ribosomal proteins andto high gradients’: the authors should discuss if this tendency is also applied to the other nervous systems.

A similar trend was observed in vertebrate retinal progenitors. We added this discussion and the reference to the text.

4. L169 ‘According to the scRNAseq data, ...after the D stage’: the increase of SoxN expression after the D stage is not found in Figs. 2A and 7A, why?

There are two peaks of SoxN transcripts according to the scRNA-seq data, one peak is before Ey stage, and the second peak is after D stage. However, SoxN protein is only expressed in the youngest NBs, correlating with the first peak of transcription. The second peak of transcription didn’t seem to result in any protein expression, likely due to post-transcriptional regulation.

5. L187 'special post-transcriptional regulation of SoxN exists': the authors should perform *in situ* hybridization to make this conclusion.

Although SoxN transcripts are present in older neuroblasts as shown by our scRNA-seq data, SoxN protein is not expressed, and we think this difference is suggestive of post-transcriptional regulation. An *in situ* hybridization experiment would only double-confirm the transcript pattern revealed by scRNA-seq. To make the conclusion that there is post-transcriptional regulation of SoxN, we would need to identify candidate post-transcriptional regulators and examine their roles on SoxN regulation, and we think these could be the focus of a future study. Therefore, we toned down our expression and did not draw this conclusion.

6. L200 'the following temporal cascade still preceded (Fig. S2C-D''')': the expression of Slp2 looks slightly accelerated.

Thanks to the reviewer for this observation. We did consistently observe that Slp2 is slightly accelerated. We have modified the text.

7. L230 'in two stripes of NBs': the two stripes of Opa expression are hard to see. The images in Fig. 3A-C should be improved, and quantifications together with Ey and Slp2 are necessary. The two layers of Opa-expressing neurons are unclear in Fig. S3A.

We improved Fig.3A-C, and quantified the relative position where these three TTFs initiate their expression in neuroblasts. In a wild-type brain, the first stripe of Opa, Ey, the second stripe of Opa and Slp are activated in the 1st, 2nd/3rd, the 3rd/4th and the 4th/5th NB, respectively. For the width of the stripes, their expression can be seen in about two, four, two, and more than two consecutive NBs, respectively (See Figure 6J, wild type).

The original Fig. S3A (current Supplementary Figure 8A) is showing a cross-sectional view, and we are now showing a triple labeling of Opa, Dpn and Slp2. The Dpn+ cells on the surface are neuroblasts, and there are two layers of Dpn- Opa+ cells which are the two layers of progeny born from the two stripes of Opa+ neuroblasts.

8. L242 'clones are not generated early enough to ...': the clones should be generated in earlier stages to confirm the results of RNAi (Fig. S3B).

We generated *opa* mutant clones several times and obtained the same phenotype: Ey and Slp are delayed. We also re-examined the phenotype when we use *vsxG4* to drive *opa* RNAi, and we see that Ey and Slp2 were either greatly delayed or lost with loss of Opa. For relatively smaller brains, we usually see loss of Ey and Slp2, but for larger brains, we can observe Ey and Slp2 activation in the oldest NBs that are located in a slightly deeper focal plane at the edge of the optic lobe (Figure 3E). We then used *ayGal4* system to drive *opa* RNAi, with which we managed to generate big clones at earlier stages, and confirmed that Ey and Slp2 are greatly delayed with loss of Opa. Therefore, we conclude that loss of Opa causes a great delay of Ey and Slp expression. Opa is required for the timely activation of Ey, but there are other TFs, such as Lola, that also have roles in activating Ey.

9. L285 'consistent with the mRNA pattern (Fig. 3I)': erm mRNA pattern should be shown.

We have referred readers to the correct figure showing the erm mRNA pattern:

"...consistent with the mRNA pattern as shown by scRNA-Seq (arrow in Figure 3J-J'", Figure 1G, Supplementary Fig. 5A)."

10. L424: the authors should describe if both of *gcm1* and *gcm2* are mutated.

We clarified in the text that in mutant clones both *gcm* and *gcm2* were mutated. Furthermore, in *gcm* RNAi clones, in which only Gcm was knocked down, we observed the same phenotype that oldest Tll+ NBs failed to exit cell cycle. This is consistent with the observation that Gcm2 is not expressed in the oldest neuroblasts. We have added the *gcm* RNAi clones to the figure5 (Figure 5D-D''').

11. L425-430: cell cycle exit should be directly tested by using general cell cycle markers such as PH3 antibody or BrdU.

Thanks for the suggestion. We tested cell cycle exit using PH3, and observed increased staining of PH3 in those oldest NBs with loss of Gcm. We added this result to our current Supplementary Figure 11 (Supplementary Fig. 11C-C'''). In addition, a lot of excessive Tll+ Dac+ neurons are produced. Therefore, we think those oldest NBs with loss of Gcm indeed could not exit cell cycle and keep producing neurons instead.

12. L436 'but may cooperate with ... (Fig. S5H)': what happens to Dap expression in UAS-*gcm* clones?

With overexpression of Gcm, Dap is induced in the clones, suggesting that Gcm is also sufficient to induce Dap expression. We added this significant result to Figure 5 (Figure 5G-G'').

13. L461-485: Southall et al. Dev. Cell 2014 (ref 80) described dedifferentiation of neurons in *lola* mutants. The *lola* RNAi phenotypes in Fig. 6D-G might be caused by neuronal dedifferentiation. The authors should discuss this issue by appropriately citing the reference. dedifferentiation of neurons

We used three RNAi lines against *lola* and observed the same phenotype: delay in temporal progression. In our RNAi clones in the larval stage at 72hr after clone induction, we only observed several ectopic NBs located at the medial edge of the clones, and we think they are probably the oldest NBs unable to exit cell cycle due to the extreme delay of the TTF cascade (Supplementary Fig. 13D). No huge increase of ectopic NBs caused by dedifferentiation of neurons was observed in our RNAi clones at this time point. In Southall et al 2014, the neuron de-differentiation phenotype is more profound at later stages (92hr after clone induction and in adult brains). Moreover, when we examine the effect of *lola*-RNAi on temporal progression, we are looking at the neuroblast surface layer. The ectopic NBs resulting from de-differentiation of neurons would be located in the deep progeny layer, therefore do not affect our observation of the neuroblast phenotype. We added the discussion of this reference.

14. L560 'Negative feedback loops with time-delay can generate striped expression patterns': clarify this

sentence.

We have added new experimental results about the regulation of Erm using Erm::V5 as a reporter (Figure 3L-M'), and have modified the discussion based on these new results.

Minor points:

Fig. 1C: the numbers inside the panel are hard to see.

We have increased the font size for these numbers.

Fig. 5D: Dap staining is hard to see. In general, dark blue is too dark, and should be changed to the other color such as light blue.

The original Fig 5D is showing loss of Dap in *gcm* mutant clones in which both *gcm* and *gcm2* are mutant. We observed the same phenotype in *gcm* RNAi clones, in which only Gcm is knocked down. We have replaced the original Fig5D with *gcm* RNAi clones (Figure 5F-F'), and used magenta for Dap, which is easier to see.

Fig. S5E: the overlap between Tll and Dac is unclear.

We improved the figure by zooming in to show the double positive cells to make it clearer. However, in wild type brains, Dac expression is usually weak in these double positive cells, and the number of Tll+Dac+ neurons is low. It is possible that in wild type brains, Tll expression is lost when Tll stage neurons mature, and Dac level is only increased after Tll is lost, while in *gcm* mutant or RNAi clones, the abnormal maintenance of Tll expression in neuroblasts may also leads to abnormal maintenance of Tll expression in neurons. This together with the increased production of Tll stage neurons enable us to see a lot of Tll+Dac+ neurons. Another possibility is that in wild type brains, Dac is only weakly expressed in neurons born at the Tll stage. In *gcm* mutant or RNAi clones, Tll is abnormally maintained in neuroblasts and neurons, and this prolonged Tll expression in neurons leads to fully activation of strong Dac expression. .

L73: Klumpfuss This has been corrected.

L98 'the gradual change of neuroblast transcriptome' This has been corrected.

L106 'but with important exceptions and complexities': this sentence is unclear. This has been clarified.

L124 'E(spl)mg::GFP that is expressed in all neuroblasts': a reference is needed. A reference has been added.

L141 'Hth, Ey, Slp1/2, D and Tll: according to Introduction, Klu should also be included. Klu mRNA is expressed widely in neuroblasts, and couldn't be used to verify the temporal trajectory (see our response to question 1).

L190: the reference #19 may be wrong. We verified #19 is the correct reference, SoxN expression was mentioned in the supplementary data of this reference.

L250 'Hth expression was expanded in both NBs and progeny (Fig. 3E, F)': is Bsh induced by the expanded Hth expression?

We did not see a clear expansion of Bsh neurons (shown below), there might be other TFs inhibiting the Bsh neuron fate.

L486: In the Nerfin-1 section, Vissers et al., Cell Rep 25, 1561, 2018 should be appropriately cited.
This reference has been properly cited.

Reviewer #2 (Remarks to the Author):

Temporal transcription factor-dependent patterning of neural stem cells allows for the generation of a wide variety of differentiated cell types during neurogenesis. This mechanism was first described in *Drosophila* brain and optic lobe neural stem cells. Accumulating evidence suggests that similar mechanisms also occur in vertebrate radial glia. Dr. Li is the first-author on a 2013 *Nature* paper that describes the temporal cascade of transcription factors that pattern neural stem cells in the *Drosophila* optic lobe. It is clear from Dr. Li's seminal study and the subsequent studies from Dr. Desplan's group that there are yet-to-be-identified genes in this cascade. Dr. Li's group took advantage of the powerful single-cell sequencing technology to define the complete temporal cascade that patterns optic lobe neural stem cells in this study.

In general, I think the scRNA-seq data the authors have generated is very rich and deeply sequenced. The investigators carried out rigorous validation of their findings by examining protein expression and genetic interactions. Aside from adding new genes to the cascade, the overall theme of the temporal cascade remains at the level of genetic interactions, identical to the Li et al., *Nature* 2013 paper. Perhaps, more in-depth analyses of the scRNA-seq analysis might reveal novel insights into the temporal cascade. For example, Pseudotime analyses shown in Fig. 1C suggest a major branchpoint. Could this branchpoint suggest some sort of heterogeneity among optic lobe neural stem cells and that more than one temporal cascades pattern optic lobe neural stem cells in a region-specific manner?

Thanks for the comments and suggestion. We think the branchpoint in the temporal trajectory is related to the cell cycle phases. As shown in Fig.1D, S phase cells are clustered on the top branch (cluster 5), while G2/M phase cells are clustered on the bottom branch (cluster 6). These two clusters express the same temporal genes: B-H1 and B-H2. Therefore, we didn't find evidence of a different temporal cascade related to the branchpoint.

Major point

1. I think their descriptions of justifications for their in silico filtering methods could be expanded upon to include more descriptions of why cells were excluded (what kinds of cells are they?).

See our responses below. Description of the excluded cells is now added to the analysis part of the Methods section.

A. I downloaded the GEO mtx counts table and processed it through MiCV (<https://micv.works>; Michki et al., bioRxiv 2020 (doi: <https://doi.org/10.1101/2020.07.02.184549>)) as follows: Downstream scRNA-seq analysis was performed using MiCV, an interactive web tool that leverages scanpy (Wolf et al., 2018) and other libraries for analysis. In brief, cells were filtered by requiring between 200-15000 unique genes/cell (to exclude debris and some doublets), and genes were filtered by requiring at least 2 cells to express it at greater than 1 UMI/cell. UMI counts were normalized to a total sum of 1000000.0 counts/cell (conversion to counts-per-million/CPM) and subsequently log-transformed by calculating $\ln(1+CPM)$ for each gene for each cell. The top 2000 highly variable genes were identified using the cell_ranger method (Zheng et al., 2017) and these genes were used to perform a principal component analysis (PCA, $n=50$ pcs). Whenever batch correction was required, data were integrated using the harmony algorithm. A neighborhood embedding was calculated with $k=20$ neighbors before finally embedding cells in a UMAP projection.

(Figure was here)

dpn expression in the starting data from GEO (some dpn+ cells are covered by dpn- cells in the plot above):

(Figure was here)

53% of all cells express dpn at a non-zero level (highlighted in plot below).

(Figure was here)

Question: what are the rest (47%) of the cells that do not express dpn? Do they also express SoxN and E(spl)mgamma? If so, why are they not MNBs? There are likely some more marker genes that can be used to explain why these cells should be excluded from analysis here.

The cells that do not express dpn also do not express mira, another marker for neuroblasts. While some of them do express SoxN and E(spl)mgamma, these are expressed at only very low levels. For example, 95% of the dpn- cells show fewer than 8 reads of SoxN and 4 reads of E(spl)mgamma. These cells are likely medulla GMCs or neurons that have some residual transcripts of SoxN and E(spl)mgamma left, and also have perdurance of fluorescent reporters.

1B. I then took out just the dpn+ cells and re-analyzed them (same protocol as above).

(Figure was here)

This leaves 6149 cells total. I believe the portion of these cells that the authors analyze are in the large chunk of cells circled below (about 55% of the dpn+cells):

(Figure was here)

Questions: Why were the other cells (top-right portion of these UMAPs) excluded from the analysis? Are they the high mito-percentage cells?

These excluded cells are indeed the high mito-percentage cells.

If so, if we consider that 10% may be too restrictive of a filtering step (there is debate around the percentage that indicates poor cell-health, and 20% is not an unheard of cutoff), I wonder a) what the mean/median percentage of mito reads is in this excluded population,

In the 3065 cells that were excluded because they had mitochondrial read percentages of 10% or higher, the median percentage was 13.78% and the mean was 16.23%.

and b) if these cells are included regardless of mito-percentage, what (if any) conclusions might change about the proposed TTF progression?

Raising the mitochondrial percentage threshold to 20% might introduce too much confounding into the analysis. First, even after regressing out mitochondrial percentage, the UMAP projection changed dramatically. In contrast to our original analysis, now most of the cells with high mitochondrial percentage cluster together by themselves. It is possible that this clustering is driven by confounding factors reflected in the high mitochondrial percentages.

Furthermore, the predicted cell cycle phases now also look very different compared to in the original analysis. There is a much smaller proportion of S phase cells, and most of them seem to be in the cluster with high mitochondrial percentage.

Is it possible that this excluded group of cells (or even some groups in the originally included set of cells) represent a separate branch in the TTF progression cascade?

It seems like most of the excluded cells express many known temporal transcription factors. However, it seems difficult to identify temporal patterns in the new group of cells.

Re-clustering the subset of cells with between 10% and 20% mitochondrial percentage is not any more informative. It is possible that the high mitochondrial percentage might be obfuscating any additional information about the TF temporal cascade carried in these cells.

Are they actually all NBs?

While these cells all express *dpn*, many of them do exhibit lower levels of other important neuroblast marker genes. We think that they are either not neuroblasts or they are neuroblasts in unhealthy conditions possibly due to the cell-dissociation and FACS sorting procedures.

Importantly, do they come from both experimental replicates, or only one of them? From my analysis here, it seems that MB_1 (the first replicate on GEO) is overrepresented in this excluded population, and that MB_2 (the second replicate) makes up the majority fraction of cells in the data included by the authors. Can the authors propose a reason for this distribution?

The high mitochondrial percentage cells are unequally distributed between the two replicates. Of the 3612 dpn+ cells from MB_1, 78.5% of the cells (2836) had mitochondrial percentages of 10% or higher. Of the 2527 dpn+ cells from MB_2, only 10% (229) had mitochondrial percentages of 10% or higher. This is possibly because that the brain dissection, cell dissociation and cell sorting process lasted longer at the first trial, and resulted in more unhealthy cells.

Do they see such a batch-variation in their own analysis, or is this a limitation of the batch-correction method used here? (Harmony, from Korsunsky et. al., 2019, Nature Methods)

We do see batch variation in our analysis even after using the batch correction method implemented in Seurat (the IntegrateData function). The left panel of the figure below shows cells with mitochondrial percentage greater than or equal to 10% while the figure on the right shows the remainder of the cells.

Furthermore, after raising the threshold to 20%, we still see batch effects even after regressing out mitochondrial percentage and then integrating data.

However, in our initial analysis where we excluded cells with mitochondrial percentages larger than 10%, the batch effect is minimized after correction using Seurat. Therefore, we will keep the 10% cut-off in our manuscript.

2. line 131-133: Cluster 8, 11, 12, and 14 appear to be outliers, but by what criteria other than UMAP position? The rationale used to exclude those cells can be improved (the marker genes expressed by those cells vs. the cells they kept should be briefly described and used for that rationale - UMAP position alone can be an indicator, but in general is not a strong enough criteria).

See response to item 2 of Reviewer 1.

3. line 137-145: Pseudotime analyses typically require a starting cell to be selected. How was this cell chosen? (typically, by high expression of 1-3 specific genes that are known to be expressed in the starting cell population; I don't believe the authors describe their criteria).

The root of the trajectory was chosen to be the vertex closest to cells with highest median *hth* expression, since *Hth* is known to be expressed in the youngest neuroblasts. This was in the method part, and we have also included this in the main text in the revision.

4. line 133-145 and Fig. 1C-D: NBs divide many times throughout their lifetime, going through the entire cell cycle each time. This is not a criticism of their analysis. The authors could attempt to regress out the effect of cell-cycle genes and re-cluster to see if their 'temporal progression' patterns were linked to progression through the cell cycle and not NB age? Alternatively, providing data that shows the fraction of cells from each cluster that are in the G1, G2/M, and S phases (as bar graphs, for instance) may help readers better understand how well-represented each phase is in each NB-type cluster/portion of the pseudotime trajectory. I believe doing a regression/showing this information will actually strengthen the conclusions that authors have made.

First of all, in addition to the bio-informatics analysis, we have experimentally verified that the TTF cascade progression is linked to the NB age. In the developing medulla, there is a neurogenesis wave sweeping from medial to lateral, and this wave sequentially converts neuroepithelial cells (NE) into medulla NBs. As a result, neuroblasts of different ages are aligned on the medial to lateral spatial axis. For example, our known TTF markers *Hth*, *Ey*, *Slp*, *D* and *Tll* are expressed in 5 stripes from lateral to medial. Our inferred pseudotime trajectory is verified by the transcript pattern of known TTFs, and we have also examined the protein expression pattern of new TTFs, and their expression pattern on the medial to lateral spatial axis are all consistent with their location on the pseudotime.

Next we reanalyzed the data according to the suggestions, and the results are included in the new Supplementary Figure 1. We calculated the fraction of cells, within each cluster found in our original analysis, in different phases of the cell cycle (Supplementary Fig. 1D). The clusters have been ordered according to the median pseudotime of the cells in each cluster, from smallest to largest. This figure suggests that there may be some relationship between our estimated pseudotime and cell cycle phase distribution, as it appears that a larger fraction of cells are in the S or G2 phases as NBs age. However, the correlation between the pseudotime and the cell cycle phase distribution is likely to be of biological significance: Cluster 13 in Fig 1C are the youngest NBs that are just transformed from neuroepithelial cells, these cells are known to be arrested at G1 phase (Orihara-Ono, et al. <https://doi.org/10.1016/j.ydbio.2010.12.044>). Cluster 2 in Fig 1C are the oldest NBs undergoing terminal divisions, so the majority of them are at G2/M phase. For the clusters in the middle, it looks like that the relative duration of G1 phase is decreasing as NBs age.

We then attempted to remove this association between pseudotime and cell cycle by regressing out cell cycle stage before clustering and estimating pseudotime. Specifically, we regressed out the S phase and G2M phase scores calculated by Seurat (*S.Score* and *G2M.Score*). Supplementary Fig. 1E and G show the resulting clusters and the expression patterns of known temporal TFs. We then estimated pseudotime. To be consistent with our original analysis, we again first identified and then removed outlier clusters. In this clustering, clusters 8, 11, 12, and 14 demonstrated the same expression patterns as the outlier clusters in our original analysis. After removing these clusters, we estimated pseudotime. However, it is clear that the estimated pseudotime is not perfectly consistent with the expression patterns of the known temporal TFs (Supplementary Fig. 1F). For instance, cluster 7 appears to have the largest

pseudotime, but cluster 1 expresses Tll, which is known to appear at later stages of NB temporal patterning.

Therefore, before visualizing the fraction of cells within each cluster in different phases of the cell cycle, we first manually decided on a cluster ordering that better matched the temporal ordering of the TFs. The result shows that there does still appear to be some association between pseudotime and cell cycle phase distribution, and G1 phase seems to be decreasing with pseudotime (Supplementary Fig. 1H) .

We also analyzed our data using the “Alternate Workflow” suggested by the Seurat developers for regressing out cell cycle. This workflow first calculates the difference between S.Score and G2M.Score and then regresses out the difference, and is supposed to be able to maintain some cell cycle effects that may be of biological significance. However, the estimated pseudotimes again were not consistent with the expression patterns of known TFs, and after manually ordering the clusters there still appeared to be an association between pseudotime and phase distribution (Supplementary Fig. 1I).

These results suggest that the association between cell cycle and NB temporal patterning may be difficult to disambiguate.

Nevertheless, our qualitative conclusions still hold true. For example, we found that many of the genes showing high correlation (both positive and negative) with pseudotime were consistent within each of the three phases; see our response to item #11 of Reviewer 3.

In summary, these results suggest that the correlation between the pseudotime and cell cycle phase distribution is likely to be of biological significance, and cannot be resolved by regressing out cell cycle. it is likely that the G1 phase is decreasing as NBs age.

5. line 184-188 and Fig. 1H: the authors used a SoxN-Gal4 line to drive expression of the reporter they used to physically enrich their cells. Is it surprising that SoxN is broadly expressed in the cells at the RNA-level? I don't think they have compelling evidence that SoxN is 'temporally varying' either, based on their RNA-seq data.

SoxN appears to have a non-linear relationship with pseudotime, as it decreases and then increases. A Spearman test for the correlation between SoxN expression and pseudotime gives a p-value of 2.5e-9. SoxN protein is only expressed in the youngest NBs, correlating with the first peak of transcription. The second peak of transcription didn't seem to result in any protein expression, likely due to post-transcriptional regulation. Since SoxN is a transcription factor, the protein level matters for its function, and the protein level does vary with time.

6. The authors propose that three temporal TFs (Dmrt99B, Hth, and SoxN) may function in parallel. Could the author exclude the possibility that Dmrt99B functions upstream of Hth and SoxN? Please examine if Dmrt99B RNAi decreases SoxN and Hth expression.

In Dmrt99B RNAi clones, SoxN or Hth expression is not affected in neuroblasts. We have added these data to (Supplementary Fig. 7K-L').

7. Is Erm expression affected in ey-mutant clones? How about Ey expression in erm-mutant clones?

We used Erm::V5 to examine how Erm expression is regulated. In ey RNAi clones, Erm expression was extended towards older NBs, and interestingly, the expression level of Erm in NBs with ey RNAi seemed to become lower compared to its wild-type stripe (Figure 3M,M'). Therefore, it is possible that a lower level of Ey at the gap between the two stripes of Opa enhances the activation of Erm, but a higher level of Ey and/or a factor induced by higher level of Ey at the second stripe of Opa represses Erm. Ey expression is decreased in erm mutant clones, but not completely gone. This result is in the current Supplementary Figure 8K,K''.

8. Opa expression patterns in erm clones appear different from ey clones in Fig. 3G & J. There appears to be one wave of Opa expression in erm-mutant clones but two waves of Opa expression in ey-mutant clones. Please clarify. If this is the case, does this result suggest Erm and Ey have different function during this transition?

Indeed, in erm mutant clones, Opa is only de-repressed in the gap between the two stripes of Opa expression. In ey RNAi clones, Opa is de-repressed in both the gap and in older neuroblasts. We also showed that Slp is required to repress the second stripe of Opa in older neuroblasts. Because Slp is lost in ey RNAi clones, but not lost in erm mutant clones, this should be the reason for the difference between the two phenotypes.

9. The authors conclude that Scro promotes the transition from the Ey stage to the Slp stage by activating Slp1 expression. Please examine Scro protein expression pattern.

We were not able to examine Scro protein expression pattern due to the lack of the antibody. However, the scro RNAi phenotype (significantly reduced Slp expression in neuroblasts) suggest that Scro is required in neuroblasts to activate Slp expression to the full level.

10. Does scroRNAi affect Erm expression?

We tested the effect of scro RNAi on Erm, and did not observe a clear change of Erm expression in neuroblasts.

11. The conclusion for Fig. 5H is overstated because the authors did not include a rescue experiment!. They should perform experiments to demonstrate Tll and Dap are the functional downstream of Gcm in regulation of gliogenesis and cell cycle exit. Otherwise, the authors should tone down their conclusion.

Indeed we haven't performed a rescue experiment, due to difficult genetic schemes. Therefore, we toned down our conclusion.

12. Other types of neuroblast showed nuclear Pros in the final division before entering quiescence or terminating the cell cycle. Multiple factors contribute to the cell cycle exit of the old neuroblast. Did you examine if Nerfin-1-positive neuroblasts show nuclear Pros in interphase?

Yes, we observed nuclear Pros in interphase in Nerfin-1-positive neuroblasts. We added the figure to the manuscript (Supplementary Figure 12A-A’’’).

13. Identification of Nerfin-1 in regulating progression and termination of the temporal cascade is interesting. Can Nerfin-1 expressing neuroblasts divide? Could they be GMCs instead? If they divide, is the division symmetric or asymmetric?

According to our scRNA-seq data, these Nerfin-1/Gcm expressing neuroblasts are at the G2/M phase, suggesting that they are undergoing the final division. They express Dpn and Mira, two markers of neuroblasts, and some of them are positive for Anti-phospho-Histone H3 (New Figure 5B-B’’’, supplementary Fig. 10B-B’’’). Based on these evidence, we think they are neuroblasts at the final G2/M phase. We examined Numb staining and were able to observe asymmetric localization of Numb in some of these final stage neuroblasts (data not shown). However, we haven’t been able to characterize whether the two daughters adopt different fates. Therefore, we were not able to draw a conclusion whether this last division is symmetric or asymmetric.

14. The authors suggest that Erm represses Opa and that Opa represses Hth. Is there any evidence that this repression cascade is direct?

These regulatory interactions are inferred from mutant phenotypes. We don’t know whether they are direct or indirect.

15. The authors suggest that Scro activates Slp. Does Scro overexpression induce higher Slp expression? Please explain the outcome of this experiment.

We tested the effect of overexpressing Scro, and found that Scro overexpression does not induce earlier or higher Slp expression (Supplementary Fig. 9A-A’’), suggesting that Scro is not sufficient to activate Slp.

16. The authors suggest that BarH1 and BarH2 activate Tll and Gcm, and that Gcm represses Tll. The rationale regarding how BarH1 and BarH2 activate gene expression with different kinetics is very unclear and difficult to imagine. Please clarify.

We have clarified this sentence: “BarH1 and BarH2 are required to activate both Tll and Gcm, but Tll is activated before Gcm, and Gcm is then required to promote gliogenesis and cell cycle exit by repressing Tll and activating Dap”. How Tll and Gcm can be activated at different time points remains unclear.

17. Data in the figure are not trivial for general audience to evaluate because of the amount of background information needed to interpret the staining patterns and the complexity of tissue architecture.

A. Please consider adding a summary illustration describing findings from previous studies and ending each figure by inserting newly obtained information into the illustration.

A scheme summarizing previous findings before this paper has been added to Supplementary Figure 1A, and each figure ends with an updated model summarizing what have been shown in this figure.

B. Please consider labeling distinct cell types and distinct waves of gene expression in the figures with arrow, arrowheads, etc, and spatially emphasize where the authors would like the reader to focus on.

We have added more arrows in the figures to emphasize where the readers should focus on.

C. Please also include the expression pattern of Gal4 drivers used in the experiment in the illustration.

We have added expression patterns of VsxGal4 and Optix Gal4 to supplementary Figure 8 (Supplementary Fig. 8F,G).

D. It is very difficult to evaluate the boundary of gene expression in multi-color images in many figures. The authors should consider including some single-color images with labeling. In Fig. 2A for example, it is very difficult to see the boundary of Ey-negative stage and -positive stage.

An Ey single-channel image is included in Fig. 2A. Single channels for Opa, Ey, Slp2 and ErmV5 are included in Figure 3.

Minor point

1. Please outline clones in 3K', 4B', 4E' 5B' 5C', 5C'', 5D' 5E'', and 5G''. These have been added.

2. in Fig1B, myGFP? This has been corrected.

3. Please clarify the rationale why Dfr expression does not require Ey if Opa is not present.

Our results suggest that Opa normally represses the generation of the first population Dfr⁺ neurons (Dfr⁺ Ap⁺ neurons), and then Erm and Ey together are required to turn off Opa at the early Ey stage to allow for the generation of Dfr⁺ Ap⁺ neurons. Thus, when Opa is knocked down by RNAi, the repression on the first population of neurons (Dfr⁺ Ap⁺ neurons) is lifted, and Ey is then no longer required for their generation. We have clarified this in the text.

4. line 23-24, "All ages" could be clearer - this scRNA-seq experiment was only performed at 1 developmental time-point.

The optic lobe medulla is a special system: a neurogenesis wave initiates in the medial edge of the OPC neuroepithelium, and spreads to the lateral direction. Due to the spreading of this wave, we can observe neuroblasts of all ages (from newly transformed to the oldest terminally dividing neuroblasts) at one developmental time-point. We emphasized this point in the introduction.

5. line 21-22, The authors say it is not known if why 'temporal progression' only happens in NBs and not in progeny. I wonder what 'temporal progression' they would expect in terminally differentiated and nonamplifying progeny, and if they mean just in MNBs or in all NBs. They point out the temporal progression of type-II INPs in [52-53].

The post-mitotic progeny (of all NBs or INPs) may inherit the TTF present in the NB at its birth time, but the inherited TTF does not go on to activate the next TTF. Our hypothesis is that some factors expressed in neurons may be preventing the TTF cascade progression, or that some other factors only present in NBs (or INPs) are required for the TTF cascade progression. What we mean is that we would like to identify those factors expressed in neurons that prevent TTF cascade progression, and those factors that are expressed in NBs to promote TTF cascade progression.

6. In Fig1B, what is the blue signal in the image of the larval brain lobe?

The Blue signal is Phalloidin staining to show the whole brain lobe. We have updated the legends.

7. Is SoxN expression affected in *hth*-mutant neuroblasts?

No. SoxN expression is not affected in *hth*-mutant neuroblasts This is now shown in Supplementary Figure 7F,F'.

8. line 203, what is the *ayGal4* system?

The *ayGal4* system uses *actin*>FRT- γ^+ -STOP-FRT-Gal4, in which *actin* promoter is driving Gal4 expression only after a STOP cassette (FRT- γ^+ -STOP-FRT) is excised by the action of *hs*>FLP. This is a way to generate clones in which Gal4 can drive the expression of UAS-transgenes. We have added information in the text.

Reviewer #3 (Remarks to the Author):

main strength

This study led by Dr. Xin Li is beautifully done and very thorough! It reveals a full set of temporal transcription factors with detailed characterization of TTF cascading, TTFs' roles in neuronal diversification, and regulation of the TTF cascade by additional factors. It provides a complete mechanistic understanding on how a single neural progenitor can consistently yield a series of multiple neuron types. This important and comprehensive work promises to be a classical paper for the field of developmental neurobiology.

Thanks for the positive comments on the significance of this study.

weakness of the study

There are no major weaknesses to this study. However, there are a few concepts that could be explained or framed differently or perhaps explored more deeply.

1. Lines 21-22 "It is also not known why temporal progression only happens in neuroblasts but not in their differentiated progeny."

and Lines 84-88: "Furthermore, temporal cascade progression only happens in neuroblasts but not in their differentiated progeny, which makes us to question whether genes functioning in the differentiation axis (NBs -> GMCs -> neurons) might play a role in regulating the temporal progression, which has never been studied before."

This premise doesn't make much sense to me. Temporal progression needs to happen only in the NB. Do you mean to say something along the lines of "It is also not known how progression of the temporal transcription factor cascade is confined only to the neuroblasts. The progeny inherit the TTF of their parent NB and do not express the subsequent TTF in the cascade."

Yes, this is exactly what we mean. We have edited the text accordingly to make it clear.

Temporal progression occurring in the post-mitotic progeny wouldn't make sense biologically (though expression of a subsequent neuronal transcription factor could be via a similar mechanism, though it would not be considered temporal progression). Thus, it is not clear what question is trying to be addressed regarding the differentiation axis regulating temporal progression.

Indeed TTF cascade progression should not happen in the post-mitotic progeny. Our hypothesis is that NB specific factors may be required for the TTF cascade progression, or neuron specific factors may prevent the TTF cascade progression. These factors must be differentially expressed along the differentiation axis (NBs vs neurons). We would like to identify these factors that are either functioning in NBs to promote TTF cascade progression, or functioning in the neurons to prevent the TTF cascade progression.

2. Lola as speed modulator:

While it is clear that Lola RNAi expands Hth expression domain, the remainder of the TTFs appear to be delayed. However, the data in Figure 6 does not clearly show expansion of the other TTF domains as 6P suggests. Using Dpn as a NB marker and counting the number of NBs expressing each TTF (comparing Lola RNAi and WT) would help make this conclusion more solid.

Thank you very much for the suggestion. We used Dpn as the marker to indicate the time of Hth, Opa, Erm, Ey and Slp's activation and compare their activation time in wild type brains and in brains with *lola* RNAi induced. For the expansion of the Hth and Opa stripes, we also counted the number of NBs within the stripe as an indication. These new data are now included in Figure 6.

3. Is Nerfin-1 actually a TTF?

Nerfin-1 in the context that is being examined is not really "functioning along the differentiation axis," rather Nerfin-1 is expressed in the NB. Indeed, Nerfin-1 may be the final TTF that drives the transition from a NB to Glioblast and cell cycle exit and. Figure 6O, suggests that Nerfin is required to turn off Tll. Do Nerfin-1 and Gcm work together (partially redundant) or is there regulation between Gcm and Nerfin-1? It would be interesting to explore the relationship between Gcm and Nerfin-1. Is Nerfin-1 required for Gcm expression or vice versa. Do BarH1/BarH2 turn on Nerfin-1? The Gcm mutant seems to have a milder phenotype than nerfin-1 RNAi with regards to excess Tll positive cells. The # of NBs that did not exit the cell cycle (rather than dedifferentiated neurons) is hard to determine based on Fig6M, perhaps a NB specific driver is needed for Nerfin-RNAi.

This is a great question. Nerfin-1 transcripts are specifically expressed in the final stage neuroblasts perfectly overlapping with that of Gcm. Loss of Nerfin-1 leads to ectopic neuroblasts, some of which express Tll and keep generating Tll+ progeny. Loss of Nerfin-1 also leads to loss of medulla neuropil glia. This suggests that Nerfin-1 might be required in final stage neuroblasts to repress Tll and promote gliogenesis, and acts as a TTF. But we realized that there is also another possibility that Nerfin-1 might be required in newly-born glia to prevent them from reverting back to neuroblasts similar to its role in newly-born neurons.

We tested whether Nerfin-1 regulates Gcm expression. In *Nerfin-1* RNAi clones, Gcm positive neuroblasts were still present, suggesting that Nerfin-1 is not required to activate Gcm. We are not able

to determine whether Gcm, BarH1 and BarH2 regulates Nerfin-1 expression due to the lack of antibodies and difficult genetics with Nerfin1-GFP as a reporter.

Finally, we would love to distinguish NBs that did not exit cell cycle from dedifferentiated neurons or glia. However, all of the drivers that are expressed in NBs are also maintained in newly-born progeny, and we don't have a good Gal80 to inhibit Gal4 activity specifically in neurons or glia progeny (the *elav>Gal80* we have starts expression already in neuroblasts). Therefore, currently, we cannot draw a definite conclusion whether Nerfin-1 is functioning as a TTF in NBs. Nevertheless, because of its specific expression pattern in final stage neuroblasts, and the possibility that it functions in neuroblasts together with Gcm to promote cell cycle exit and gliogenesis, in the revised version we put the Nerfin-1 part together with the Gcm part, but we also pointed out the other possibility that Nerfin-1 may function in newly born glia, and that we are not sure whether Nerfin-1 is acting as a TTF in NBs.

minor comments:

1. Line 36: When you say "neural stem cells" I assume that you mean vertebrate, mammalian or human neural stem cells.

We changed neural stem cells to neural progenitors or neuroblasts.

2. Lines 50-51: "Postembryonic NBs, including larval ventral nerve cord, central brain and optic lobe NBs, also employ temporal cascades that are yet different from the embryonic one." "Temporal cascades" may be too broad...postembryonic CB NBs don't really employ a "cascade" for temporal patterning, as "cascade" implies a series of TTFs driving the subsequent TTF expression.

We apologize for the inaccurate summary. We have corrected this sentence to: " Postembryonic NBs, including larval ventral nerve cord, central brain and optic lobe NBs, also undergo temporal patterning dependent on either TTF cascades or opposing temporal gradients of two RNA binding proteins."

3. Lines 71-72: "Among them, Hth, Ey, Slp and D are each required for the expression of the corresponding neuronal transcription factors to control neural fates." This is a little bit vague as written—can you clarify what is meant by "corresponding neuronal transcription factors" and "neural fates"?

We changed to a more specific statement: Among them, Hth, Ey, Slp and D are each required for the generation of neurons expressing Brain-specific homeobox (Bsh), Drifter (Dfr), Twin of Eyeless (Toy) /Sox102F, or Ets at 65A (Ets65a), respectively.

4. Lines 82-83: "As a previous TTF is only necessary but not sufficient to activate the next TTF in medulla NBs" This information needs a reference.

A Reference has been added.

5. Lines 98-99 "We observed growth and metabolism related genes showed high to low gradients". This is grammatically awkward. There are 2 subjects and 2 verbs "we observed" and "genes showed"

We corrected this sentence to: "We observed high to low gradients for growth and metabolism related genes, and low to high gradients for genes involved in gene expression regulation and neural differentiation as neuroblasts age."

6. Figure 1B: FACS sorting: It looks like the PE-A laser settings may have been too high (there seem to be many cells above 105) and many Red/Green double+ neuroblasts may have been missed. Did you also confirm NBs were larger (forward scatter)?

That's true. When we were doing the sorting, we wanted to make sure to select double positive ones, so we set the threshold higher. There could have been double positive cells missed. However, since we dissected ~120 brains, we were able to get sufficient number of cells to be sequenced. Yes, we did confirm that NBs are larger.

7. Lines 127-128: "After quality control and filtering for neuroblasts, our data contained 3074 cells expressing between 261 and 6409 genes, with a median of 3682 expressed genes per cell." According to the methods (lines 730-731) you sequenced over 11K cells. Were most cells lost during QC or filtering for Dpn? What % of sequenced cells were excluded with QC or the NB filtering step?

See response to item 1 of Reviewer 2.

8. Cluster 8 seems to be rather large for an outlier. Is there any clue what the "outlier" clusters could be? Could they be NBs from a different brain region? Or a mixture of cell types (rather than single cells).

See response to item 2 of Reviewer 1 and item 1 of Reviewer 2.

9. Figure 1F: There is a lot of overlap in the expression of the known TTFs. Does this RNAseq analysis align well with the previously published antibody data? It would be helpful to see a figure demonstrating some of the overlap in expression of TTFs. This would also help show where a TTF may be missing—for instance it appears there are cells in cluster 0 that do not express hth or ey and there seems to be a gap between D and Tll expression clusters 5&6.

We plotted pairs of TFs in UMAP plots (Figure below) as suggested. Consistent with our previous antibody data, transcripts of "neighboring" TTFs (Ey with Slp, and Slp with D) show significant overlaps in neuroblasts (see Supplementary Fig. 4B-D). These pair-plots also clearly revealed gaps between Hth and Ey, D and Tll, and after the Tll stage, indicative of missing TTFs (see Supplementary Fig. 4A,E). Pair-plotting between D, newly-identified BarH genes, and Tll showed that B-H genes fill the gap between D and Tll (see Supplementary Fig. 5B-D). We have provided these figures as Supplementary figures, and incorporated the discussion into the text.

10. Lines 149-151: references to figure are missing "Genes encoding ribosomal proteins and metabolism enzymes showed significant high to low gradients (Figure S1A), while genes involved in gene expression regulation and neural development showed low to high gradients (Figure S1B)".

We have added the references to figures in these places.

11. Do the genes that show opposite expression gradients (lines 147-154 and Fig S1) correlate with cell cycle rather than simply temporal patterning? This is important to consider as the ratio of cells captured in different cell cycle stages correlates with temporal patterning and many of the GO terms are related to cell cycle processes. If controlled for cell cycle stage, would the gradients still persist? For instance, if

only cells that were in G1 were assessed over pseudo time, would the expression gradients remain? We repeated our gradient analyses within each cell cycle phase. In each phase, there was a significant overlap between the top 200 genes showing positive correlation with pseudotime in the phase, and the top 200 genes from our original analysis. The same was true for the top 200 genes showing negative correlation. The following table shows that there was more than 50% overlap in each of the lists. We included these new analysis in supplementary figure 3.

	Overlap in top 200 genes positively correlated with pseudotime	Overlap in top 200 genes negatively correlated with pseudotime
G1 phase	129	102
S phase	135	164
G2M phase	158	133

12. Lines 163-165 “We next examined the expression of these TFs at the protein level by immunostaining of 3rd instar larval brains using available antibodies or GFP-fusion lines.” This data is not shown. It would be helpful to have a table that shows the antibodies/GFP-fusion lines etc. that were tested, and which showed temporal differences.

From supplementary Figure 2C, we selected those that show clear temporal patterns, so we didn’t need to do a very extensive screening. We added a table as you suggested. Please see Supplementary Table 5.

13. Lines 179-182 “For a few other candidate TFs with temporal expression patterns, including Oaz, Hbn, Scrt and Sba (Figure 1H), we either did not observe an effect on the temporal progression (Oaz, Sba, Scrt) using available RNAi lines, or we are lacking effective reagents (Hbn). Therefore, we did not include these TFs in further analysis.”

This data is not shown. Can it be included in the supplement

We have added these data to Supplementary Figure 6 (Supplementary Fig. 6A-C’).

14. The fact that SoxN protein has a clear temporal pattern but the transcript less so (temporal post transcriptional regulation) suggests that other potential TTF’s transcripts may not be differentially expressed. Are there temporally regulated RNA binding proteins?

We obtained a list of RNA binding proteins from Flybase. We then tested their expression patterns for Spearman correlation with the inferred pseudotime and found that most of them were significantly associated with pseudotime at the FDR < 0.05 level. We have provided these heatmaps in Supplementary Figure 6 (Supplementary Fig. 6D-E).

15. Since SoxN is post-transcriptionally regulated, the data suggests that Ey (or an Ey downstream gene) is responsible for the post-transcriptional regulation of SoxN.

Yes, that is what we also speculated. However, we only looked at transcription factors for this paper. We need to also examine other genes to understand the mechanism in a separate study.

16. Lines 197-200: The conclusion that SoxN “cooperates with Hth to specify the Bsh neuron fate” should come after Hth expression is described as “unaffected” in SoxN mutants

We have changed the text accordingly.

17. Figure 3 A-C' It is clear that all of these images are from the same sample. The colocalization of red and blue are not as clear as with red/magenta and green. For B and C, it would be helpful to have 3 color panels with Dpn shown to highlight the NBs.

We improved the figure as you suggested, with 3 color panels with Dpn labeling added to Figure 3B-C''.

18. The alternating stipes of Erm and Opa proteins are very interesting, suggesting potential cross regulation. Does Erm inhibit Opa expression in the neural epithelial cells? Does Opa inhibit Erm? I later discovered that this was covered in the discussion. Perhaps this sentence (lines 557-558) "Due to the lack of Erm antibody, we haven't examined how Erm expression is regulated" could be moved to the results section.

In *erm* mutant clones, *Opa* expression is de-repressed in the gap between the two stripes, but not de-repressed in neural-epithelial cells, suggesting that Erm does not inhibit *Opa* expression in NE. Using *Erm::V5* line, we were able to examine whether *Opa* or *Ey* regulates *Erm* expression. With loss of *Opa*, *Erm::V5* expression is lost, suggesting *Opa* is required for the activation of *Erm*. With loss of *Ey*, *Erm::V5* is expanded into older neuroblasts, but the level becomes weaker, suggesting that a low level of *Ey* might initially enhance the activation of *Erm* by *Opa*, but later a higher level of *Ey* and/or other factors activated by higher level of *Ey* are responsible for repressing *Erm*. We have included these new data in Figure 3 (Figure 3L-M').

19. Interestingly, *Erm* and *Opa* transcripts do not appear to have such clear alternating expression as the antibody/transgene. Could there be post-transcriptional regulation at play?

Our scRNA seq data only included NB (*Dpn*+) cells, so the *Erm* stripe in NE cannot be seen. In the pair-plotting for *Erm* and *Opa* below, we do see *Erm* transcripts (green) are present in the gap between the two groups of *Opa* (red) expressing neuroblasts. In the NBs that have *Erm* transcripts (green), *Opa* transcripts are not present or reduced. The *Erm* stripe only has a small number of cells, maybe this is the reason why the gap between the two *Opa* stripes is not that striking. Nevertheless, it is still possible that *Opa* is under specific post-transcriptional regulation, which we do not have experimental evidence to support at this point.

20. Figure 6H: Can Dpn, Nerfin and Gcm triple staining be performed? Are there antibodies available for Nerfin-1 or Gcm?

Unfortunately, antibodies for Gcm and Nerfin-1 are not available. Instead, we co-stained Dpn, β -galactosidase and GFP after combining *gcm-LacZ* (BDSC 5445) with Nerfin-1-GFP. And showed that β -galactosidase and Nerfin-1 can be co-expressed in oldest NBs. One concern is that the expression of β -galactosidase might be a bit earlier than that of Gcm, reflecting the transcriptional pattern instead of the translational pattern of Gcm.

Reviewed by Tzumin & Rosa

REVIEWERS' COMMENTS

Reviewer #1 (Remarks to the Author):

I think the authors addressed all of my concerns and the revised manuscript was significantly improved.

About the description of Klu, a recent paper describes that Klu expression is under the control of Notch signaling and is not expressed in E(spl)my::GFP-negative cells (Nat Comm 12, 2083, 2021). If so, it is not surprising if E(spl)my::GFP-sorted scRNA-seq does not detect the change in Klu expression. The authors should briefly mention this point.

Reviewer #2 (Remarks to the Author):

The authors made excellent effort to satisfactorily address the reviewer's critiques. I fully support publication of this study.

Reviewer #3 (Remarks to the Author):

Zhu and colleagues have made a noteworthy effort to respond to reviewers' concerns. This impressive and thorough manuscript will be an important addition to the field.

We request the authors reconsider how the Lola data is introduced and discussed.

The justification of examining Lola or other genes along the "differentiation axis" is intriguing but has not been accomplished in this paper. With Nerfin-1 and Lola, they have looked at two genes that have been previously shown to be involved in maintaining differentiation of neurons and examined their NB specific roles in temporal patterning. As such, We do not believe the results provide "clues as to why the temporal progression of TTFs only proceeds in neuroblasts [Line 119]."

The background within the "Lola, functioning along the differentiation axis, participate in the temporal cascade regulation" section makes a reader anticipate isoform-specific knock down of lola and/or lola knock down specifically in GMCs and neurons followed by testing whether TTFs now progress in the neurons. The data itself which suggests NB expressed Lola regulates the speed of the TTF cascade is intriguing and more emphasis could be placed on that aspect (discussion of speed regulation of genetic cascades etc.), rather than the differentiation axis.

The Slp expressing domain seems to be smaller in Fig 6I. This may be due to all the TTFs being expanded (thus shifted) and the ability to only assess the 1st nine NBs. Another possibility is that early TTFs are expanded at the expense of late TTFs. Can these ideas be discussed?

Minor points (suggestions to improve understanding of the text):

"Might" is used 24 times in the text. The use of "might" suggests that something is less likely to happen (it is only one of many possibilities). Better choices depending on the situation are: "may" or "is/are likely" or "likely verb."

Lines 69-73: This should be broken up into two sentences. The first sentence can clarify that required generation of the neurons makes Hth, Ey Slp D TTFs. The second sentence can say that it is not clear whether Klu is a TTF and why.

Lines 139-143: This section still makes a reader pause and wonder about the "outliers". Perhaps if you rewrite this section to first describe how and why you identify the other clusters as medulla NE and NBs and then explain why 8,11,12,14 don't fit into those criteria, your point will come across clearer.

Lines 217-227 consist of a single sentence separated by semicolons. Please consider breaking up into separate sentences for each thought.

Line 240: "Our scRNA-seq analysis did reveal a larger number of RNA-binding proteins..." Larger should be "large" or "high" unless you are comparing the number of RNA-binding proteins to the number of another type of gene.

Line 273: please clarify that neuronal Bsh expression was lost

Line 522-523: "As Tll is not required for glia production suggesting that Tll is not required to activate Gcm, there should be another TTF directly upstream of Gcm and required for Gcm expression." This sentence is grammatically awkward.

Line 534: Nerin-1 should be Nerfin-1

Line 624: Please also add erm and ey expressing NBs are also expanded in numbers.

Reviewer #1 (Remarks to the Author):

I think the authors addressed all of my concerns and the revised manuscript was significantly improved.

Thanks for the comment.

About the description of Klu, a recent paper describes that Klu expression is under the control of Notch signaling and is not expressed in E(spl)my::GFP-negative cells (Nat Comm 12, 2083, 2021). If so, it is not surprising if E(spl)my::GFP-sorted scRNA-seq does not detect the change in Klu expression. The authors should briefly mention this point.

Thanks for the suggestion. We have mentioned this point and added this reference to the text.

Reviewer #2 (Remarks to the Author):

The authors made excellent effort to satisfactorily address the reviewer's critiques. I fully support publication of this study.

Thanks for the support.

Reviewer #3 (Remarks to the Author):

Zhu and colleagues have made a noteworthy effort to respond to reviewers' concerns. This impressive and thorough manuscript will be an important addition to the field.

Thanks for the comment.

We request the authors reconsider how the Lola data is introduced and discussed.

The justification of examining Lola or other genes along the "differentiation axis" is intriguing but has not been accomplished in this paper. With Nerfin-1 and Lola, they have looked at two genes that have been previously shown to be involved in maintaining differentiation of neurons and examined their NB specific roles in temporal patterning. As such, We do not believe the results provide "clues as to why the temporal progression of TTFs only proceeds in neuroblasts [Line 119]."

We agree with the reviewers' comment, and we have removed the statement about our results providing "clues as to why the temporal progression of TTFs only proceeds in neuroblasts".

The background within the "Lola, functioning along the differentiation axis, participate in the temporal cascade regulation" section makes a reader anticipate isoform-specific knock down of lola and/or lola knock down specifically in GMCs and neurons followed by testing

whether TTFs now progress in the neurons. The data itself which suggests NB expressed Lola regulates the speed of the TTF cascade is intriguing and more emphasis could be placed on that aspect (discussion of speed regulation of genetic cascades etc.), rather than the differentiation axis.

Thanks for the reviewers' suggestion. We have now changed the way that the Lola part is introduced and discussed. We removed all discussion about the differentiation axis, and focused on how the speed of the temporal cascade is regulated, as suggested by the reviewers.

The Slp expressing domain seems to be smaller in Fig 6l. This may be due to all the TTFs being expanded (thus shifted) and the ability to only assess the 1st nine NBs. Another possibility is that early TTFs are expanded at the expense of late TTFs. Can these ideas be discussed?

The Slp expressing domain appears to be smaller, and this is because we could only reliably assess the first nine NBs. Older NBs are dislocated into deep layers, and it is hard to determine their relative age. Therefore, it is not clear whether the Slp stage is expanded or not. We have added this discussion to the text.

Minor points (suggestions to improve understanding of the text):

"Might" is used 24 times in the text. The use of "might" suggests that something is less likely to happen (it is only one of many possibilities). Better choices depending on the situation are: "may" or "is/are likely" or "likely verb."

We have changed "might" to "may" or "is/are likely" or "likely verb" depending on the context.

Lines 69-73: This should be broken up into two sentences. The first sentence can clarify that required generation of the neurons makes Hth, Ey Slp D TTFs. The second sentence can say that it is not clear whether Klu is a TTF and why.

This long sentence is now broken into two sentences.

Lines 139-143: This section still makes a reader pause and wonder about the "outliers". Perhaps if you rewrite this section to first describe how and why you identify the other clusters as medulla NE and NBs and then explain why 8,11,12,14 don't fit into those criteria, your point will come across clearer.

We have changed the order of description according to this suggestion: markers expressed by the main body of neuroblasts first, and then outliers.

Lines 217-227 consist of a single sentence separated by semicolons. Please consider breaking up into separate sentences for each thought.

This very long sentence is now broken into several sentences.

Line 240: "Our scRNA-seq analysis did reveal a larger number of RNA-binding proteins..." Larger should be "large" or "high" unless you are comparing the number of RNA-binding proteins to the number of another type of gene.

We have corrected this typo.

Line 273: please clarify that neuronal Bsh expression was lost

We have clarified that it is the neuronal Bsh expression that was lost.

Line 522-523: "As Tll is not required for glia production suggesting that Tll is not required to activate Gcm, there should be another TTF directly upstream of Gcm and required for Gcm expression." This sentence is grammatically awkward.

We have broken this sentence into two sentences, and modified the grammar.

Line 534: Nerin-1 should be Nerfin-1

We have corrected this typo.

Line 624: Please also add erm and ey expressing NBs are also expanded in numbers.

We have added this statement. Thanks!

By Tzumin & Rosa